# When is Momentum Extragradient Optimal?
# A Polynomial-Based Analysis

**Junhyung Lyle Kim**[*]                                                    *jlylekim@rice.edu*
*Rice University, Department of Computer Science*

**Gauthier Gidel**                                                    *gauthier.gidel@umontreal.ca*
*Université de Montréal, Department of Computer Science and Operations Research*
*Mila – Quebec AI Institute, Canada CIFAR AI Chair*

**Anastasios Kyrillidis**                                                    *anastasios@rice.edu*
*Rice University, Department of Computer Science*

**Fabian Pedregosa**                                                    *pedregosa@google.com*
*Google DeepMind*

**Reviewed on OpenReview:** *https://openreview.net/forum?id=ZLVbQEu4Ab*

## Abstract

The extragradient method has gained popularity due to its robust convergence properties for differentiable games. Unlike single-objective optimization, game dynamics involve complex interactions reflected by the eigenvalues of the game vector field's Jacobian scattered across the complex plane. This complexity can cause the simple gradient method to diverge, even for bilinear games, while the extragradient method achieves convergence. Building on the recently proven accelerated convergence of the momentum extragradient method for bilinear games (Azizian et al., 2020b), we use a polynomial-based analysis to identify three distinct scenarios where this method exhibits further accelerated convergence. These scenarios encompass situations where the eigenvalues reside on the (positive) real line, lie on the real line alongside complex conjugates, or exist solely as complex conjugates. Furthermore, we derive the hyperparameters for each scenario that achieve the fastest convergence rate.

## 1 Introduction

While most machine learning problems are formulated as minimization problems, a growing number of works rely instead on game formulations that involve multiple players and objectives. Examples of such problems include generative adversarial networks (GANs) (Goodfellow et al., 2014), actor-critic algorithms (Pfau & Vinyals, 2016), sharpness aware minimization (Foret et al., 2021), and fine-tuning language models from human feedback (Munos et al., 2023). This increasing interest in game formulations motivates further theoretical exploration of differentiable games.

Optimizing differentiable games presents challenges absent in minimization problems due to the interplay of multiple players and objectives. Notably, the game Jacobian's eigenvalues are distributed on the complex plane, exhibiting richer dynamics compared to single-objective minimization, where the Hessian eigenvalues are restricted to the real line. Consequently, even for simple bilinear games, standard algorithms like the gradient method fail to converge (Mescheder et al., 2018; Balduzzi et al., 2018; Gidel et al., 2019).

Fortunately, the extragradient method (EG), originally introduced by Korpelevich Korpelevich (1976), offers a solution. Unlike the gradient method, EG demonstrably converges for bilinear games (Tseng, 1995). This

---

[*]Authors after JLK are listed in alphabetical order.
This paper extends Kim et al. (2022) presented at the NeurIPS 2022 Optimization for Machine Learning Workshop.

has sparked extensive research analyzing EG from different perspectives, including variational inequality (Gidel et al., 2018; Gorbunov et al., 2022), stochastic (Li et al., 2021), and distributed (Liu et al., 2020; Beznosikov et al., 2021) settings.

Most existing works, including those mentioned earlier, analyze EG and relevant algorithms by assuming some structure on the objectives, such as (strong) monotonicity or Lipschitzness (Solodov & Svaiter, 1999; Tseng, 1995; Daskalakis & Panageas, 2018; Ryu et al., 2019; Azizian et al., 2020a). Such assumptions, in the context of differentiable games, confine the distribution of the eigenvalues of the game Jacobian; for instance, strong monotonicity implies a lower bound on the real part of the eigenvalues, and the Lipschitz assumption implies an upper bound on the magnitude of the eigenvalues of the Jacobian.

Building upon the limitations of prior assumptions, Azizian et al. (2020b) showed that the key factor for effectively analyzing game dynamics lies in the spectrum of the Jacobian on the complex plane. Through a polynomial-based analysis, they demonstrated that first-order methods can sometimes achieve faster rates using momentum. This is achieved by replacing the smoothness and monotonicity assumptions with more precise assumptions on the distribution of the Jacobian eigenvalues, represented by simple shapes like ellipses or line segments. Notably, Azizian et al. (2020b) proved that for bilinear games, the extragradient method with momentum achieves an accelerated convergence rate.

In this work, we take a different approach by asking the *reverse question*: for what shapes of the Jacobian spectrum does the momentum extragradient (MEG) method achieve optimal performance? This reverse analysis allows us to study the behavior of MEG in specific settings depending on the hyperparameter setup, encompassing:

- *Minimization*, where all Jacobian eigenvalues lie on the positive real line.

- *Regularized bilinear games*, where all eigenvalues are complex conjugates.

- *Intermediate case*, where eigenvalues are both on the real line and as complex conjugates (illustrated in Figure 1).

Our contributions can be summarized as follows:

- **Characterizing MEG convergence modes**: We derive the residual polynomials of MEG for affine game vector fields and identify three distinct convergence modes based on hyperparameter settings. This analysis can then be applied to different eigenvalue structures of the Jacobian (see Theorem 3).

- **Optimal hyperparameters and convergence rates**: For each eigenvalue structure, we derive the optimal hyperparameters of MEG and its (asymptotic) convergence rates. For minimization, MEG exhibits "super-acceleration," where a constant improvement upon classical lower bound rate is attained,[1] similarly to the gradient method with momentum (GDM) with cyclical step sizes (Goujaud et al., 2022). For the other two cases involving imaginary eigenvalues, MEG exhibits accelerated convergence rates with the derived optimal hyperparameters.

- **Comparison with other methods**. We compare MEG's convergence rates with gradient (GD), GDM, and extragradient (EG) methods. For the considered game classes, none of these methods achieve (asymptotically) accelerated rates (Corollaries 1 and 2), unlike MEG. In Section 7, we validate our findings through numerical experiments, including scenarios with slight deviations from our initial assumptions.

## 2 Problem Setup and Related Work

Following Letcher et al. (2019); Balduzzi et al. (2018), we define the $n$-player differentiable game as a family of twice continuously differentiable losses $\ell_i : \mathbb{R}^d \to \mathbb{R}$, for $i = 1, \ldots, n$. The player $i$ controls the parameter $w^{(i)} \in \mathbb{R}^{d_i}$. We denote the concatenated parameters by $w = [w^{(1)}, \ldots, w^{(n)}] \in \mathbb{R}^d$, where $d = \sum_{i=1}^n d_i$.

---

[1]Note that achieving this improvement is possible by having additional information beyond just the largest (smoothness) and smallest (strong convexity) eigenvalues of the Jacobian.

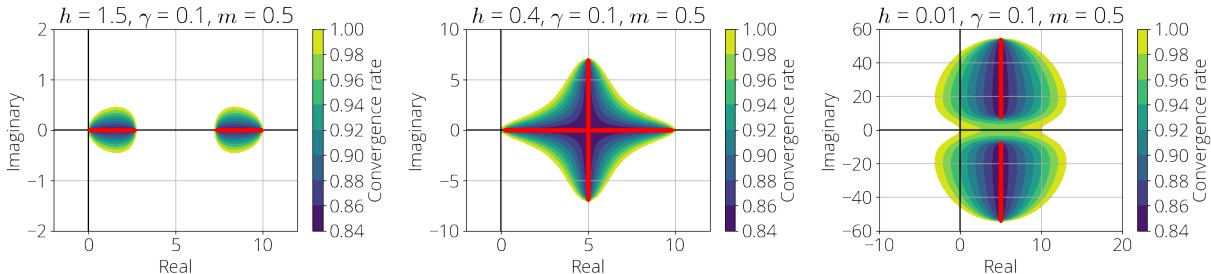

Figure 1: *Convergence rates of MEG in terms of the game Jacobian eigenvalues.* The step sizes for MEG, $h$ and $\gamma$, and the momentum parameter $m$ are set up according to each case of Theorem 3, illustrating three distinct convergence modes of MEG. For each case, the red line indicates the robust region (c.f., Definition 1) where MEG achieves the optimal convergence rate.

For this problem, a Nash equilibrium satisfies: $w^{(i)^\star} \in \arg\min_{w^{(i)} \in \mathbb{R}^{d_i}} \ell_i\left(w^{(i)}, w^{(\neg i)^\star}\right) \quad \forall i \in \{1, \ldots, n\}$, where the notation $\cdot^{(\neg i)}$ denotes all indices except for $i$. We also define the vector field $v$ of the game as the concatenation of the individual gradients: $v(w) = [\nabla_{w^{(1)}}\ell_1(w) \cdots \nabla_{w^{(n)}}\ell_n(w)]^\top$, and denote its associated Jacobian with $\nabla v$.

Unfortunately, finding Nash equilibria for general games remains an *intractable problem* (Shoham & Leyton-Brown, 2008; Letcher et al., 2019).[2] Therefore, instead of directly searching for Nash equilibria, we focus on finding *stationary points* of the game's vector field $v$. This approach is motivated by the fact that any Nash equilibrium necessarily corresponds to a stationary point of the gradient dynamics. In other words, we aim to solve the following problem:

$$\text{Find} \quad w^\star \in \mathbb{R}^d \quad \text{such that} \quad v(w^\star) = 0. \tag{1}$$

**Notation.** $\Re(z)$ and $\Im(z)$ respectively denote the real and the imaginary part of a complex number $z$. The spectrum of a matrix $M$ is denoted by $\mathrm{Sp}(M)$, and its spectral radius by $\rho(M) := \max\{|\lambda| : \lambda \in \mathrm{Sp}(M)\}$. $M \succ 0$ denotes that $M$ is a positive-definite matrix. $\mathbb{C}_+$ denotes the complex plane with positive real part, and $\mathbb{R}_+$ denotes positive real numbers.

## 2.1 Related Work

The extragradient method, originally introduced in Korpelevich (1976), is a popular algorithm for solving (unconstrained) variational inequality problems in (1) (Gidel et al., 2018). There are several works that study the convergence rate of EG for (strongly) monotone problems (Tseng, 1995; Solodov & Svaiter, 1999; Nemirovski, 2004; Monteiro & Svaiter, 2010; Mokhtari et al., 2020; Gorbunov et al., 2022). Under similar settings, stochastic variants of EG are studied in Palaniappan & Bach (2016); Hsieh et al. (2019; 2020); Li et al. (2021). However, as mentioned earlier, assumptions like (strong) monotonicity or Lipchtizness may not accurately represent *how* Jacobian eigenvalues are distributed.

Instead, we make more fine-grained assumptions on these eigenvalues, to obtain the optimal hyperparameters and convergence rates for MEG via polynomial-based analysis. Such analysis dates back to the development of the conjugate gradient method (Hestenes & Stiefel, 1952), and is still actively used; for instance, to derive lower bounds (Arjevani & Shamir, 2016), to develop accelerated decentralized algorithms (Berthier et al., 2020), and to analyze average-case performance (Pedregosa & Scieur, 2020; Domingo-Enrich et al., 2021).

On that end, we use the following lemma (Chihara, 2011), which elucidates the connection between first-order methods and (residual) polynomials, when the vector field $v$ is affine. First-order methods are the ones in which the sequence of iterates $w_t$ is in the span of previous gradients: $w_t \in w_0 + \mathrm{span}\{v(w_0), \ldots, v(w_{t-1})\}$.

---

[2]Formulating Nash equilibrium search as a nonlinear complementarity problem makes it inherently difficult, classified as PPAD-hard (Daskalakis et al., 2009; Letcher et al., 2019).

**Lemma 1** (Chihara (2011)). *Let $w_t$ be the iterate generated by a first-order method after $t$ iterations, with $v(w) = Aw + b$. Then, there exists a real polynomial $P_t$, of degree at most $t$, satisfying:*

$$w_t - w^\star = P_t(A)(w_0 - w^\star), \tag{2}$$

*where $P_t(0) = 1$, and $v(w^\star) = Aw^\star + b = 0$.*

By taking $\ell_2$-norms, (2) further implies the following worst-case convergence rate:

$$\|w_t - w^\star\| = \|P_t(A)(w_0 - w^\star)\| \leqslant \|P_t(Z\Lambda Z^{-1})\| \cdot \|w_0 - w^\star\| \leqslant \sup_{\lambda \in \mathcal{S}^\star} |P_t(\lambda)| \cdot \|Z\| \|Z^{-1}\| \cdot \|w_0 - w^\star\|, \tag{3}$$

where $A = Z\Lambda Z^{-1}$ is the diagonalization of $A$,[3] and the constant $\|Z\| \|Z^{-1}\|$ disappears if $A$ is a normal matrix. Hence, the worst-case convergence rate of a first-order method can be analyzed by studying the associated residual polynomial $P_t$ evaluated at the eigenvalues $\lambda$ of the Jacobian $\nabla v = A$, distributed over the set $S^\star$.

**Unlocking Faster Rates Through Fine-Grained Spectral Shapes.** While Azizian et al. (2020b) characterized lower bounds and optimality for certain first-order methods under simple spectral shapes, we posit that a more granular understanding of $\mathcal{S}^\star$ could unlock even faster convergence rates. By meticulously analyzing the residual polynomials of MEG, we identify specific spectral shapes where MEG exhibits optimal performance. This approach resonates with recent advancements in optimization literature (Oymak, 2021; Goujaud et al., 2022), which demonstrate that knowledge beyond merely the largest and smallest eigenvalues (i.e., smoothness and strong convexity) can lead to accelerated convergence in convex smooth minimization.

## 3 Momentum Extragradient via Chebyshev Polynomials

In this section, we delve into the intricate dynamics of the momentum extragradient method (MEG) by harnessing the power of residual polynomials and Chebyshev polynomials.

MEG iterates according to the following update rule:

$$\text{(MEG)} \quad w_{t+1} = w_t - hv(w_t - \gamma v(w_t)) + m(w_t - w_{t-1}), \tag{4}$$

where $h$ is the step size, $\gamma$ is the extrapolation step size, and $m$ is the momentum parameter.

The extragradient method (EG), which serves as the foundation for MEG, was originally proposed by Korpelevich (1976) for saddle point problems. It has garnered renewed interest due to its remarkable ability to converge in certain differentiable games, such as bilinear games, where the standard gradient method falters (Gidel et al., 2019; Azizian et al., 2020b;a).

For completeness, we remind the gradient method with momentum (GDM):

$$\text{(GDM)} \quad w_{t+1} = w_t - hv(w_t) + m(w_t - w_{t-1}), \tag{5}$$

from which the gradient method (GD) can be obtained by setting $m = 0$.

As a first-order method (Arjevani & Shamir, 2016; Azizian et al., 2020b), MEG's behavior can be elegantly analyzed through the lens of residual polynomials, as established in Lemma 1. The following theorem unveils the specific residual polynomials associated with MEG:

**Theorem 1** (Residual polynomials of MEG and their Chebyshev representation). *Consider the momentum extragradient method (MEG) in (4) with a vector field of the form $v(w) = Aw + b$. The residual polynomials associated with MEG can be expressed as follows:*

$$\tilde{P}_0(\lambda) = 1, \quad \tilde{P}_1(\lambda) = 1 - \frac{h\lambda(1 - \gamma\lambda)}{1 + m}, \quad \text{and} \quad \tilde{P}_{t+1}(\lambda) = (1 + m - h\lambda(1 - \gamma\lambda))\tilde{P}_t(\lambda) - m\tilde{P}_{t-1}(\lambda).$$

---

[3]Note that almost all matrices are diagonalizable over $\mathbb{C}$, in the sense that the set of non-diagonalizable matrices has Lebesgue measure zero (Hetzel et al., 2007).

*Remarkably, these polynomials can be elegantly rewritten in terms of Chebyshev polynomials of the first and second kind, denoted by $T_t(\cdot)$ and $U_t(\cdot)$, respectively:*

$$P_t^{MEG}(\lambda) = m^{t/2}\left(\tfrac{2m}{1+m}T_t(\sigma(\lambda)) + \tfrac{1-m}{1+m}U_t(\sigma(\lambda))\right), \; where \; \sigma(\lambda) \equiv \sigma(\lambda; h, \gamma, m) = \frac{1+m-h\lambda(1-\gamma\lambda)}{2\sqrt{m}}. \quad (6)$$

*The term $\sigma(\lambda)$, which encapsulates the interplay between step sizes, momentum, and eigenvalues, is referred to as the link function.*

The residual polynomials of MEG and GDM, intriguingly, share a similar structure but differ in their link functions. Below are the residual polynomials of GDM, expressed in Chebyshev polynomials (Pedregosa, 2020):

$$P_t^{\mathrm{GDM}}(\lambda) = m^{t/2}\left(\tfrac{2m}{1+m}T_t(\xi(\lambda)) + \tfrac{1-m}{1+m}U_t(\xi(\lambda))\right), \quad where \quad \xi(\lambda) = \frac{1+m-h\lambda}{2\sqrt{m}}. \quad (7)$$

Notice that the residual polynomials of MEG in (6) and that of GDM in (7) are identical, except for the link functions $\sigma(\lambda)$ and $\xi(\lambda)$, which enter as arguments in $T_t(\cdot)$ and $U_t(\cdot)$.

The differences in these link functions are paramount because the behavior of Chebyshev polynomials hinges decisively on their argument's domain:

**Lemma 2** (Goujaud & Pedregosa (2022)). *Let $z$ be a complex number, and let $T_t(\cdot)$ and $U_t(\cdot)$ be the Chebyshev polynomials of the first and second kind, respectively. The sequence $\left\{\left|\tfrac{2m}{1+m}T_t(z) + \tfrac{1-m}{1+m}U_t(z)\right|\right\}_{t\geqslant 0}$ grows exponentially in $t$ for $z \notin [-1, 1]$, while for $z \in [-1, 1]$, the following bounds hold:*

$$|T_t(z)| \leqslant 1 \quad and \quad |U_t(z)| \leqslant t + 1. \quad (8)$$

Therefore, to study the optimal convergence behavior of MEG, we are interested in the case where the set of step sizes and the momentum parameters lead to $|\sigma(\lambda; h, \gamma, m)| \leqslant 1$ so that we can use the bounds in (8). We will refer to those sets of eigenvalues and hyperparameters as the *robust region*, as defined below.

**Definition 1** (Robust region of MEG). *Consider the MEG method in (4) expressed via Chebyshev polynomials, as in (6). We define the set of eigenvalues and hyperparameters such that the image of the link function $\sigma(\lambda; h, \gamma, m)$ lies in the interval $[-1, 1]$ as the **robust region**, and denote it with $\sigma^{-1}([-1, 1])$.*

Although polynomial-based analysis requires the assumption that the vector field is affine, it captures intuitive insights into how various algorithms behave in different settings, as we remark below.

**Remark 1.** *From the definition of $\xi(\lambda)$ in (7), one can infer why negative momentum can help the convergence of GDM (Gidel et al., 2019) when $\lambda \in \mathbb{R}_+$: it forces GDM to stay within the robust region, $|\xi(\lambda)| \leqslant 1$. One can also infer the divergence of GDM in the presence of complex eigenvalues, unless, for instance, complex momentum is used (Lorraine et al., 2022). Similarly, the residual polynomial of GD is $P_t^{GD}(\lambda) = (1 - h\lambda)^t$ (Goujaud & Pedregosa, 2022, Example 4.2), and can easily diverge in the presence of complex eigenvalues, which can potentially be alleviated by using complex step sizes. On the contrary, thanks to the quadratic link function of MEG in (6), it can converge for much wider subsets of complex eigenvalues.*

By analyzing the residual polynomials of MEG, we can also characterize the asymptotic convergence rate of MEG for any combination of hyperparameters, as summarized in the next theorem.

**Theorem 2** (Asymptotic convergence rate of MEG). *Suppose $v(w) = Aw + b$. The asymptotic convergence rate of MEG in (4) is:[4]*

$$\limsup_{t\to\infty} \sqrt[2t]{\frac{\|w_t - w^\star\|}{\|w_0 - w^\star\|}} = \begin{cases} \sqrt[4]{m}, & if \;\; \bar\sigma \leqslant 1 \;\; (robust\ region); \\ \sqrt[4]{m}\left(\bar\sigma + \sqrt{\bar\sigma^2 - 1}\right)^{1/2}, & if \;\; \bar\sigma \in \left(1, \tfrac{1+m}{2\sqrt{m}}\right); \\ \geqslant 1 \;(no\ convergence), & otherwise, \end{cases} \quad (9)$$

*where $\bar\sigma = \sup_{\lambda\in\mathcal{S}^\star} |\sigma(\lambda; h, \gamma, m)|$, and $\sigma(\lambda; h, \gamma, m) \equiv \sigma(\lambda)$ is the link function of MEG defined in (6).*

---

[4]The reason why we take the $2t$-th root is to normalize by the number of vector field computations; we compare in Section 4 the asymptotic rate of MEG in (9) with other gradient methods that use a single vector field computation in the recurrences, such as GD and GDM.

Optimal hyperparameters for MEG that we obtain in Section 4 minimize the asymptotic convergence rate above. Note that the optimal hyperparameters vary based on the set $\mathcal{S}^\star$, which we detail in Section 3.2.

### 3.1 Three Modes of the Momentum Extragradient

Within the robust region of MEG, we can compute its worst-case convergence rate based on (3) as follows:

$$\sup_{\lambda \in \mathcal{S}^\star} |P_t^{\mathrm{MEG}}(\lambda)| \overset{(6)}{\leqslant} m^{t/2}\Big(\tfrac{2m}{1+m} \sup_{\lambda \in \mathcal{S}^\star} |T_t(\sigma(\lambda))| + \tfrac{1-m}{1+m} \sup_{\lambda \in \mathcal{S}^\star} |U_t(\sigma(\lambda))|\Big)$$

$$\overset{(8)}{\leqslant} m^{t/2}\Big(\tfrac{2m}{1+m} + \tfrac{1-m}{1+m}(t+1)\Big) \leqslant m^{t/2}(t+1). \tag{10}$$

Since the Chebyshev polynomial expressions of MEG in (6) and that of GDM[5] are identical except for the link functions, the convergence rate in (10) applies to both MEG and GDM, as long as the link functions $|\sigma(\lambda)|$ and $|\xi(\lambda)|$ are bounded by 1. As a result, we see that the asymptotic convergence rate in (9) only depends on the momentum parameter $m$, when the hyperparameters are restricted to the robust region. This fact was utilized in tuning GDM for strongly convex quadratic minimization (Zhang & Mitliagkas, 2019).

The robust region of MEG can be described with the four extreme points below (derivation in the appendix):

$$\sigma^{-1}(-1) = \tfrac{1}{2\gamma} \pm \sqrt{\tfrac{1}{4\gamma^2} - \tfrac{(1+\sqrt{m})^2}{h\gamma}}, \quad \text{and} \quad \sigma^{-1}(1) = \tfrac{1}{2\gamma} \pm \sqrt{\tfrac{1}{4\gamma^2} - \tfrac{(1-\sqrt{m})^2}{h\gamma}}. \tag{11}$$

The above four points and their intermediate values characterize the set of Jacobian eigenvalues $\lambda$ that can be mapped to $[-1, 1]$. The distribution of these eigenvalues can vary in three different modes depending on the selected hyperparameters of MEG, as stated in the following theorem.

**Theorem 3.** *Consider the momentum extragradient method in (4), expressed with the Chebyshev polynomials as in (6). Then, the robust region (c.f., Definition 1) have the following three modes:*

- ***Case 1:*** *If $\frac{h}{4\gamma} \geqslant (1+\sqrt{m})^2$, then $\sigma^{-1}(-1)$ and $\sigma^{-1}(1)$ are all real numbers;*

- ***Case 2:*** *If $(1-\sqrt{m})^2 \leqslant \frac{h}{4\gamma} < (1+\sqrt{m})^2$, then $\sigma^{-1}(-1)$ are complex, and $\sigma^{-1}(1)$ are real;*

- ***Case 3:*** *If $(1-\sqrt{m})^2 > \frac{h}{4\gamma}$, then $\sigma^{-1}(-1)$ and $\sigma^{-1}(1)$ are all complex numbers.*

**Remark 2.** *Theorem 3 offers guidance on how to set up the hyperparameters for MEG. This depends on the Jacobian spectra of the game problem being considered. For instance, if one observes only real eigenvalues (i.e., the problem is in fact minimization), the main step size $h$ should be at least $4\times$ larger than the extrapolation step size $\gamma$, based on the condition $\frac{h}{4\gamma} \geqslant (1+\sqrt{m})^2$.*

We illustrate Theorem 3 in Figure 1. We first set the hyperparameters according to each condition in Theorem 3. We then discretize the interval $[-1, 1]$, and plot $\sigma^{-1}([-1, 1])$ for each case, represented by red lines. We can see the quadratic link function induced by MEG allows interesting eigenvalue dynamics to be mapped onto the $[-1, 1]$ segment, such as the cross-shape observed in Case 2. Moreover, although MEG exhibits the best rates within the robust region, it does not necessarily diverge outside of it, as in the second case of Theorem 2. We illustrate the convergence region of MEG measured by $\sqrt[2t]{|P_t(\lambda)|} < 1$ from (6) for $t = 2000$, with different colors indicating varying convergence rates, which slow down as one moves away from the robust region. Interestingly, Figure 1 (right) shows that MEG can also converge in the absence of monotonicity (i.e., in the presence of Jacobian eigenvalues with negative real part) (Gorbunov et al., 2023).

### 3.2 Robust Region-Induced Problem Cases

We classify problem classes into three distinct cases based on Theorem 3, each reflecting a different mode of the robust region (Figure 2):

---

[5]Asymptotically, GDM enjoys $\sqrt{m}$ convergence rate instead of the $\sqrt[4]{m}$ of MEG, as it uses a single vector field computation per iteration instead of the two. However, these are not directly comparable, as the values of $m$ that correspond to the robust region are not the same.

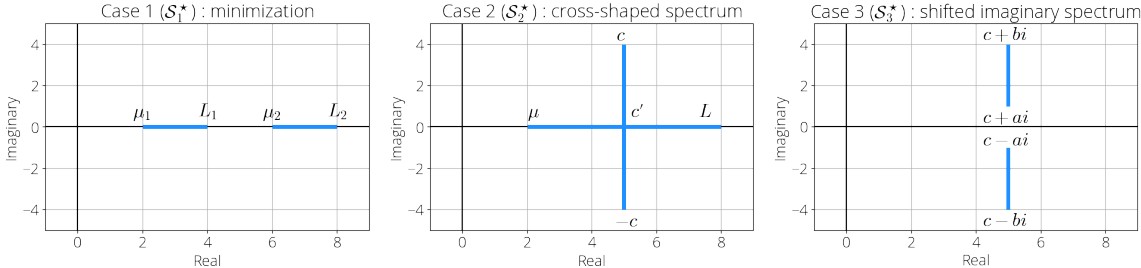

Figure 2: *Illustration of the three spectrum models where MEG achieves accelerated convergence rates.*

**Case 1:** The problem reduces to minimization, where the Jacobian eigenvalues are distributed on the (positive) real line, but as a *union* of two intervals. We can model such spectrum as:

$$\text{Sp}(\nabla v) \subset \mathcal{S}_1^\star = [\mu_1, L_1] \cup [\mu_2, L_2] \subset \mathbb{R}_+. \tag{12}$$

Above generalizes the Hessian spectrum that arise in minimizing $\mu$-strongly convex and $L$-smooth functions, i.e. $\lambda \in [\mu, L]$. This spectrum can be obtained from (12) by setting $\mu_1 = \mu$, $L_2 = L$, and $L_1 = \mu_2$. It was empirically observed that, during DNN training, sometimes a few eigenvalues of the Hessian have significantly larger magnitudes (Papyan, 2020). In such cases, (12) can be more precise than a single interval $[\mu, L]$. In particular, Goujaud et al. (2022) utilized (12), and showed that the GDM with alternating step sizes can achieve a (constant factor) improvement over the traditional lower bound for strongly convex and smooth quadratic objectives.

In Section 4, we show that MEG enjoys similar improvement. To show that, we define the following quantities following Goujaud et al. (2022), which will be used to obtain the convergence rate of MEG in (18) for this problem class.

$$\zeta := \frac{L_2 + \mu_1}{L_2 - \mu_1} = \frac{1 + \tau}{1 - \tau}, \quad \text{and} \quad R := \frac{\mu_2 - L_1}{L_2 - \mu_1} \in [0, 1). \tag{13}$$

Here, $\zeta$ is the ratio between the center of $\mathcal{S}_1^\star$ and its radius, and $\tau := L_2/\mu_1$ is the inverse condition number. $R$ is the relative gap of $\mu_2 - L_1$ and $L_2 - \mu_1$, which becomes 0 if $\mu_2 = L_1$ (i.e., $\mathcal{S}_1^\star$ becomes $[\mu_1, L_2]$).

**Case 2:** In this case, the Jacobian eigenvalues are distributed both on the real line and as complex conjugates, exhibiting a *cross-shaped* spectrum. We model this spectrum as:

$$\text{Sp}(\nabla v) \subset \mathcal{S}_2^\star = [\mu, L] \cup \{z \in \mathbb{C} : \mathfrak{R}(z) = c' > 0, \ \mathfrak{I}(z) \in [-c, c]\}. \tag{14}$$

The first set $[\mu, L]$ denotes a segment on the real line, reminiscent of the Hessian spectrum for minimizing $\mu$-strongly convex and $L$-smooth functions. The second set has a fixed real component ($c' > 0$), along with imaginary components symmetric across the real line (i.e., complex conjugates), as the Jacobian is real.

This is a strict generalization of the purely imaginary interval $\pm[ai, bi]$ commonly considered in the bilinear games literature (Liang & Stokes, 2019; Azizian et al., 2020b; Mokhtari et al., 2020). While many recent papers on bilinear games cite GANs (Goodfellow et al., 2014) as a motivation, the work in Berard et al. (2020, Figure 4) empirically shows that the spectrum of GANs is not contained in the imaginary axis; the cross-shaped spectrum model above might be closer to some of the observed GAN spectra.

**Case 3:** In this case, the Jacobain eigenvalues are distributed only as complex conjugates, with a fixed real component, exhibiting a *shifted imaginary* spectrum. We model this spectrum as:

$$\text{Sp}(\nabla v) \subset \mathcal{S}_3^\star = [c + ai, c + bi] \cup [c - ai, c - bi] \subset \mathbb{C}_+. \tag{15}$$

Again, (15) generalizes bilinear games, where the spectrum reduces to $\pm[ai, bi]$ with $c = 0$.

**Examples of Cases 2 and 3 in quadratic games.** To understand these spectra better, we provide examples using quadratic games. Consider the following two player quadratic game, where $x \in \mathbb{R}^{d_1}$ and $y \in \mathbb{R}^{d_2}$ are the parameters controlled by each player, whose loss functions respectively are:

$$\ell_1(x,y) = \frac{1}{2}x^\top S_1 x + x^\top M_{12} y + x^\top b_1 \quad \text{and} \quad \ell_2(x,y) = \frac{1}{2}y^\top S_2 y + y^\top M_{21} x + y^\top b_2, \tag{16}$$

where $S_1, S_2 \succ 0$. Then, the vector field can be written as:

$$v(x,y) = \begin{bmatrix} S_1 x + M_{12} y + b_1 \\ M_{21} x + S_2 y + b_2 \end{bmatrix} = Aw + b, \text{ where } A = \begin{bmatrix} S_1 & M_{12} \\ M_{21} & S_2 \end{bmatrix}, \ w = \begin{bmatrix} x \\ y \end{bmatrix}, \text{ and } b = \begin{bmatrix} b_1 \\ b_2 \end{bmatrix}. \tag{17}$$

If $S_1 = S_2 = 0$ and $M_{12} = -M_{21}^\top$, the game Jacobian $\nabla v = A$ has only purely imaginary eigenvalues (Azizian et al., 2020b, Lemma 7), recovering bilinear games.

As the second and the third spectrum models in (14) and (15) generalize bilinear games, we can consider more complex quadratic games, where $S_1$ and $S_2$ does not have to be 0. Specifically, when $M_{12} = -M_{21}^\top$, and they share common bases with $S_1$ and $S_2$ specified in the below proposition, then $\mathrm{Sp}(A)$ has a cross-shaped spectrum in (14) of Case 2 and a shifted imaginary spectrum in (15) of Case 3.

**Proposition 1.** *Let $A$ be a matrix of the form $\begin{bmatrix} S_1 & B \\ -B^\top & S_2 \end{bmatrix}$, where $S_1, S_2 \succ 0$. Without loss of generality, assume that $dim(S_1) > dim(S_2) = d$. Then,*

- *Case 2: $\mathrm{Sp}(A)$ has a cross-shape if there exist orthonormal matrices $U, V$ and diagonal matrices $D_1, D_2$ such that $S_1 = U\, diag(a, \ldots, a, D_1)U^\top$, $S_2 = V\, diag(a, \ldots, a)V^\top$, and $B = UD_2V^\top$.*

- *Case 3: $\mathrm{Sp}(A)$ has a shifted imaginary shape if there exist orthonormal matrices $U, V$ and diagonal matrix $D_2$ such that $S_1 = U\, diag(a, \ldots, a)U^\top$, $S_2 = V\, diag(a, \ldots, a)V^\top$, and $B = UD_2V^\top$.*

We can interpret Case 3 as a *regularized* bilinear game, where $S_1$ and $S_2$ are diagonal matrices with a constant eigenvalue. This implies that the players cannot control their parameter $x$ and $y$ arbitrarily, which can be seen in the loss functions in (16), where $S_1$ and $S_2$ appears in terms $x^\top S_1 x$ and $y^\top S_2 y$. Case 2 can be interpreted similarly, but player 1 (without loss of generality) has more flexibility in its parameter choice due to the additional diagonal matrix $D_1$ in the eigenvalue decomposition of $S_1$.

## 4 Optimal Parameters and Convergence Rates

In this section, we obtain the optimal hyperparameters of MEG (in the sense that they achieve the fastest asymptotic convergence rate), for each spectrum model discussed in the previous section.

**Case 1: minimization.** When the condition in Case 1 of Theorem 3 holds (i.e., $\frac{h}{4\gamma} \geqslant (1 + \sqrt{m})^2$), both $\sigma^{-1}(-1)$ and $\sigma^{-1}(1)$ (and their intermediate values), line on the real line, forming a union of two intervals (see Figure 1, left). The robust region in this case, denoted $\sigma^{-1}_{\mathrm{Case}_1}([-1,1])$, is expressed as:

$$\left[\frac{1}{2\gamma} - \sqrt{\frac{1}{4\gamma^2} - \frac{(1-\sqrt{m})^2}{h\gamma}}, \frac{1}{2\gamma} - \sqrt{\frac{1}{4\gamma^2} - \frac{(1+\sqrt{m})^2}{h\gamma}}\right] \bigcup \left[\frac{1}{2\gamma} + \sqrt{\frac{1}{4\gamma^2} - \frac{(1+\sqrt{m})^2}{h\gamma}}, \frac{1}{2\gamma} + \sqrt{\frac{1}{4\gamma^2} - \frac{(1-\sqrt{m})^2}{h\gamma}}\right] \subset \mathbb{R}_+.$$

For this case, the optimal hyperparameters of MEG in terms of the worst-case asymptotic convergence rate in (9) can be set as below.

**Theorem 4** (Case 1). *Consider solving (1) for games where the Jacobian has the spectrum in (12). For this problem, the optimal hyperparameters for the momentum extragradient method in (4) are:*

$$h = \frac{4(\mu_1 + L_2)}{(\sqrt{\mu_2 + L_1} + \sqrt{\mu_1 + L_2})^2}, \ \gamma = \frac{1}{\mu_1 + L_2} = \frac{1}{\mu_2 + L_1}, \quad \text{and} \quad m = \left(\frac{\sqrt{\mu_2 L_1} - \sqrt{\mu_1 L_2}}{\sqrt{\mu_2 L_1} + \sqrt{\mu_1 L_2}}\right)^2 = \left(\frac{\sqrt{\zeta^2 - R^2} - \sqrt{\zeta^2 - 1}}{\sqrt{\zeta^2 - R^2} + \sqrt{\zeta^2 - 1}}\right)^2.$$

Recalling (9), we immediately get the asymptotic convergence rate from Theorem 4. Further, this formula can be simplified in the ill-conditioned regime, where the inverse condition number $\tau := \mu_1/L_2 \to 0$:

$$\sqrt[4]{m} = \left( \frac{\sqrt{\zeta^2 - R^2} - \sqrt{\zeta^2 - 1}}{\sqrt{\zeta^2 - R^2} + \sqrt{\zeta^2 - 1}} \right)^{1/2} \underset{\tau \to 0}{=} 1 - \frac{2\sqrt{\tau}}{\sqrt{1 - R^2}} + o(\sqrt{\tau}). \tag{18}$$

From (18), we see that MEG achieves an accelerated convergence rate $1 - O(\sqrt{\tau})$, which is known to be "optimal" for this function class, and can be asymptotically achieved by GDM[6] (Polyak, 1987) (see also Theorem 8 with $\theta = 1$). Surprisingly, this rate can be further improved by the factor $\sqrt{1 - R^2}$, exhibiting "super-acceleration" phenomenon enjoyed by GDM with (optimal) cyclical step sizes (Goujaud et al., 2022). Note that achieving this improvement is possible by having additional information beyond just the largest $(L_2)$ and smallest $(\mu_1)$ eigenvalues of the Hessian.

**Case 2: cross-shaped spectrum.** If the condition in Case 2 of Theorem 3 is satisfied (i.e., $(1 - \sqrt{m})^2 \leqslant \frac{h}{4\gamma} < (1 + \sqrt{m})^2$), then $\sigma^{-1}(-1)$ are complex, while $\sigma^{-1}(1)$ are real (c.f., Figure 1, middle). We can write the robust region $\sigma^{-1}_{\text{Case}_2}([-1, 1])$ as:

$$\underbrace{\left[ \frac{1}{2\gamma} - \sqrt{\frac{1}{4\gamma^2} - \frac{(1 - \sqrt{m})^2}{h\gamma}}, \frac{1}{2\gamma} + \sqrt{\frac{1}{4\gamma^2} - \frac{(1 - \sqrt{m})^2}{h\gamma}} \right]}_{\subset \mathbb{R}_+} \bigcup \underbrace{\left[ \frac{1}{2\gamma} - \sqrt{\frac{1}{4\gamma^2} - \frac{(1 + \sqrt{m})^2}{h\gamma}}, \frac{1}{2\gamma} + \sqrt{\frac{1}{4\gamma^2} - \frac{(1 + \sqrt{m})^2}{h\gamma}} \right]}_{\subset \mathbb{C}_+}.$$

Here, the first interval lies on $\mathbb{R}_+$, as the square root term is real; conversely, in the second interval, the square root term is imaginary, with the fixed real component: $\frac{1}{2\gamma}$. We summarize the optimal hyperparameters for this case in the next theorem.

**Theorem 5** (Case 2). *Consider solving* (1) *for games where the Jacobian has a cross-shaped spectrum as in* (14). *For this problem, the optimal hyperparameters for the momentum extragradient method in* (4) *are:*

$$h = \frac{16(\mu + L)}{(\sqrt{4c^2 + (\mu + L)^2} + \sqrt{4\mu L})^2}, \quad \gamma = \frac{1}{\mu + L}, \quad \text{and} \quad m = \left( \frac{\sqrt{4c^2 + (\mu + L)^2} - \sqrt{4\mu L}}{\sqrt{4c^2 + (\mu + L)^2} + \sqrt{4\mu L}} \right)^2.$$

We get the asymptotic rate from Theorem 5, which simplifies in the ill-conditioned regime $\tau := \mu/L \to 0$ as:

$$\sqrt[4]{m} = \left( \frac{\sqrt{4c^2 + (\mu + L)^2} - \sqrt{4\mu L}}{\sqrt{4c^2 + (\mu + L)^2} + \sqrt{4\mu L}} \right)^{1/2} \underset{\tau \to 0}{=} 1 - \frac{2\sqrt{\tau}}{\sqrt{(2c/L)^2 + 1}} + o(\sqrt{\tau}). \tag{19}$$

We see that MEG achieves accelerated convergence rate $1 - O(\sqrt{\mu/L})$, as long as $c = O(L)$. We remark that this rate is optimal in the following sense. The lower bound for the problems with cross-shaped spectrum in (14) must be slower than the existing ones for minimizing $\mu$-strongly convex and $L$-smooth functions, as the former is strictly more general. Since we reach the same asymptotic optimal rate, this must be optimal.

**Case 3: shifted imaginary spectrum.** Lastly, if the condition in Case 3 of Theorem 3 is satisfied (i.e., $\frac{h}{4\gamma} < (1 - \sqrt{m})^2$), then $\sigma^{-1}(-1)$ and $\sigma^{-1}(1)$ (and the intermediate values) are all complex conjugates (c.f., Figure 1, right). We can write the robust region $\sigma^{-1}_{\text{Case}_3}([-1, 1])$ as:

$$\left[ \frac{1}{2\gamma} + \sqrt{\frac{1}{4\gamma^2} - \frac{(1 + \sqrt{m})^2}{h\gamma}}, \frac{1}{2\gamma} + \sqrt{\frac{1}{4\gamma^2} - \frac{(1 - \sqrt{m})^2}{h\gamma}} \right] \bigcup \left[ \frac{1}{2\gamma} - \sqrt{\frac{1}{4\gamma^2} - \frac{(1 - \sqrt{m})^2}{h\gamma}}, \frac{1}{2\gamma} - \sqrt{\frac{1}{4\gamma^2} - \frac{(1 + \sqrt{m})^2}{h\gamma}} \right] \subset \mathbb{C}_+.$$

We modeled such spectrum in (15), which generalizes bilinear games, where the spectrum reduces to $\pm[ai, bi]$ (i.e., with $c = 0$). We summarize the optimal hyperparameters for this case below.

---

[6]Precisely, GDM with optimal step size and momentum asymptotically achieve $1 - 2\sqrt{\tau} + o(\sqrt{\tau})$ convergence rate, as $\tau \to 0$ (Goujaud & Pedregosa, 2022, Proposition 3.3).

**Theorem 6** (Case 3)**.** *Consider solving* (1) *for games where the Jacobian has a shifted imaginary spectrum in* (15)*. For this problem, the optimal hyperparameters for the momentum extragradient method in* (4) *are:*

$$h = \frac{8c}{(\sqrt{c^2+a^2}+\sqrt{c^2+b^2})^2}, \quad \gamma = \frac{1}{2c}, \quad and \quad m = \left(\frac{\sqrt{c^2+b^2}-\sqrt{c^2+a^2}}{\sqrt{c^2+b^2}+\sqrt{c^2+a^2}}\right)^2.$$

Similarly to before, we compute the asymptotic convergence rate from Theorem 4 using (9).

$$\sqrt[4]{m} = \left(\frac{\sqrt{c^2+b^2}-\sqrt{c^2+a^2}}{\sqrt{c^2+b^2}+\sqrt{c^2+a^2}}\right)^{1/2} = \left(1 - \frac{2\sqrt{c^2+a^2}}{\sqrt{c^2+b^2}+\sqrt{c^2+a^2}}\right)^{1/2} \tag{20}$$

Note that by setting $c = 0$, the rate in (20) matches the lower bound of bilinear game: $\sqrt{\frac{b-a}{b+a}}$ (Azizian et al., 2020b, Proposition 5). Further, with $c > 0$, the convergence rate in (20) improves, highlighting the contrast between vanilla bilinear games and their regularized counterpart.

**Remark 3.** *Notice that the optimal momentum $m$ in both Theorems 5 and 6 are positive. This is in contrast to Gidel et al. (2019), where the **gradient** method with negative momentum is studied. This difference elucidates distinct dynamics of how momentum interacts with the **gradient** and the **extragradient** methods.*

## 5 Comparison with Other Methods

Having established MEG's asymptotic convergence rates for various spectrum models, we now compare it with other first-order methods, including GD, GDM, and EG.

**Comparison with GD and EG.** Building upon the fixed-point iteration framework established by Polyak (1987), Azizian et al. (2020a) interpret both GD and EG as fixed-point iterations. Within this framework, iterates of a method are generated according to:

$$w_{t+1} = F(w_t), \quad \forall t \geqslant 0, \tag{21}$$

where $F : \mathbb{R}^d \to \mathbb{R}^d$ is an operator representing the method. However, analyzing this scheme in general settings poses challenges due to the potential nonlinearity of $F$.

To address this, under conditions of twice differentiability of $F$ and proximity of $w$ to the stationary point $w^\star$, the analysis can be simplified by linearizing $F$ around $w^\star$:

$$F(w) \approx F(w^\star) + \nabla F(w^\star)(w - w^\star).$$

Then, for $w_0$ in a neighborhood of $w^\star$, one can obtain an asymptotic convergence rate of (21) by studying the spectral radius of the Jacobian at the solution: $\rho(\nabla F(w^\star)) \leqslant \rho^\star < 1$. This implies that (21) locally converges linearly to $w^\star$ at the rate $O((\rho^\star + \varepsilon)^t)$ for $\varepsilon \geqslant 0$. Further, if $F$ is linear, $\varepsilon = 0$ (Polyak, 1987).

The corresponding fixed point operators $F_h^{\mathrm{GD}}$ and $F_h^{\mathrm{EG}}$ of GD and EG[7] respectively are:

$$\text{(GD)} \quad w_{t+1} = w_t - hv(w_t) = F_h^{\mathrm{GD}}(w_t), \quad \text{and} \tag{22}$$

$$\text{(EG)} \quad w_{t+1} = w_t - hv(w_t - hv(w_t)) = F_h^{\mathrm{EG}}(w_t). \tag{23}$$

The local convergence rate can then be obtained by bounding the spectral radius of the Jacobian of the operators under certain assumptions. We summarize the relevant results below.

**Theorem 7** (Azizian et al. (2020a); Gidel et al. (2019))**.** *Let $w^\star$ be a stationary point of $v$. Further, assume the eigenvalues of $\nabla v(w^\star)$ all have positive real parts. Then, denoting $\mathcal{S}^\star := Sp(\nabla v(w^\star))$,*

*1. For the gradient method in* (22) *with step size $h = \min_{\lambda \in \mathcal{S}^\star} \mathfrak{R}(1/\lambda)$, it satisfies:[8]*

$$\rho(\nabla F_h^{GD}(w^\star))^2 \leqslant 1 - \min_{\lambda \in \mathcal{S}^\star} \mathfrak{R}\left(\frac{1}{\lambda}\right) \min_{\lambda \in \mathcal{S}^\star} \mathfrak{R}(\lambda). \tag{24}$$

---

[7]Azizian et al. (2020a) assumes that EG uses the same step size $h$ for both the main and the extrapolation steps.
[8]Note that the spectral radius $\rho$ is squared, but asymptotically is almost the same as $\sqrt{1-x} \leqslant 1 - x/2$.

2. *For the extragradient method in* (23) *with step size* $h = (4 \sup_{\lambda \in \mathcal{S}^\star} |\lambda|)^{-1}$, *it satisfies:*

$$\rho(\nabla F_h^{EG}(w^\star))^2 \leqslant 1 - \tfrac{1}{4}\left(\frac{\min_{\lambda \in \mathcal{S}^\star} \mathfrak{R}(\lambda)}{\sup_{\lambda \in \mathcal{S}^\star} |\lambda|} + \frac{1}{16}\frac{\min_{\lambda \in \mathcal{S}^\star} |\lambda|^2}{\sup_{\lambda \in \mathcal{S}^\star} |\lambda|^2}\right). \tag{25}$$

We can determine the convergence rate of GD and EG by using Theorem 7 since all three cases of our spectrum models in (12), (14), and (15) meet the condition that the eigenvalues of $\nabla v(w^\star)$ have positive real parts. The following corollary summarizes this result.

**Corollary 1.** *With the conditions in Theorem 7, for each case of the Jacobian spectrum $\mathcal{S}_1^\star$, $\mathcal{S}_2^\star$, and $\mathcal{S}_3^\star$, the gradient method in* (22) *and the extragradient method in* (23) *satisfy the following:*

- **Case 1:** $Sp(\nabla v) \subset \mathcal{S}_1^\star = [\mu_1, L_1] \cup [\mu_2, L_2] \in \mathbb{R}_+$:

$$\rho(\nabla F_h^{GD}(w^\star))^2 \leqslant 1 - \tfrac{\mu_1}{L_2}, \quad and \quad \rho(\nabla F_h^{EG}(w^\star))^2 \leqslant 1 - \tfrac{1}{4}\left(\tfrac{\mu_1}{L_2} + \tfrac{\mu_1^2}{16L_2^2}\right). \tag{26}$$

- **Case 2:** $Sp(\nabla v) \subset \mathcal{S}_2^\star = [\mu, L] \cup \{z \in \mathbb{C} : \mathfrak{R}(z) = c' > 0,\ \mathfrak{I}(z) \in [-c, c]\}$:

$$\rho(\nabla F_h^{GD}(w^\star))^2 \leqslant \begin{cases} 1 - \frac{2\mu}{4c^2/(L-\mu)+(L-\mu)} & if\ c \geqslant \sqrt{\frac{L^2-\mu^2}{4}}, \\ 1 - \frac{\mu}{L} & otherwise. \end{cases} \tag{27}$$

$$\rho(\nabla F_h^{EG}(w^\star))^2 \leqslant \begin{cases} 1 - \tfrac{1}{4}\left(\frac{\mu}{\sqrt{c^2+((L-\mu)/2)^2}} + \frac{\mu^2}{16(c^2+((L-\mu)/2)^2)}\right) & if\ c \geqslant \sqrt{\frac{3L^2+2L\mu-\mu^2}{4}}, \\ 1 - \tfrac{1}{4}\left(\frac{\mu}{L} + \frac{\mu^2}{16L^2}\right) & otherwise. \end{cases}$$

- **Case 3:** $Sp(\nabla v) \subset \mathcal{S}_3^\star = [c + ai, c + bi] \cup [c - ai, c - bi] \in \mathbb{C}_+$:

$$\rho(\nabla F_h^{GD}(w^\star))^2 \leqslant 1 - \tfrac{c^2}{c^2+b^2}, \quad and \quad \rho(\nabla F_h^{EG}(w^\star))^2 \leqslant 1 - \tfrac{1}{4}\left(\frac{c}{\sqrt{c^2+b^2}} + \frac{c^2+a^2}{16(c^2+b^2)}\right). \tag{28}$$

In Case 1, we see from (26) that both GD and EG have convergence rates $1 - O(\mu_1/L_2) = 1 - O(\tau)$. MEG, on the other hand, has an accelerated convergence rate of $1 - O(\sqrt{\tau})$, as well as an additional constant improvement by a factor of $\sqrt{1 - R^2}$, as we showed in (18). Moving on to Case 2, we showed in (19) that MEG enjoys an accelerated convergence rate of $1 - O(\sqrt{\mu/L})$ as long as $c = O(L)$. However, both GD and EG in (27) have non-accelerated convergence under the same condition. Lastly, for Case 3, we showed in (20) that MEG achieves an asymptotic rate that matches the known lower bound for bilinear games: $\sqrt{\frac{b-a}{b+a}}$, with $c = 0$; further, the rate of MEG improves if $c > 0$. On the contrary, GD and EG suffer from slower rates, as shown in (28).

**Comparison with GDM.** We now compare the convergence rate of MEG with that of GDM, which iterates as in (5). In Azizian et al. (2020b), it was shown that GD is the optimal method for games where the Jacobian eigenvalues are within a *disc* in the complex plane. This suggests that acceleration is not possible for this type of problem.[9] On the other hand, it is well-known that GDM achieves an accelerated convergence rate for strongly-convex (quadratic) minimization, where the eigenvalues of the Hessian lie on the (strictly positive) real line segment (Polyak, 1987). Hence, Azizian et al. (2020b) studies the intermediate case, where the Jacobian eigenvalues are within an ellipse, which can be thought of as the real segment $[\mu, L]$ perturbed with $\epsilon$ in an elliptic way. That is, they consider the spectral shape:[10]

$$K_\epsilon = \left\{z \in \mathbb{C} : \left(\frac{\mathfrak{R}z-(\mu+L)/2}{(L-\mu)/2}\right)^2 + \left(\frac{\mathfrak{I}z}{\epsilon}\right)^2 \leqslant 1\right\}.$$

Similarly to GD and EG above, in Azizian et al. (2020b), GDM is interpreted as a fixed point iteration:[11]

$$w_{t+1} = w_t - hv(w_t) + m(w_t - w_{t-1}) = F^{GDM}(w_t, w_{t-1}). \tag{29}$$

To study the convergence rate of GDM, we use the following theorem from Azizian et al. (2020b):

---

[9]Yet, one can consider the case where, e.g., a cross-shape is contained in a disc. Then, by knowing more fine-grained structure of the Jacobian spectrum, MEG can have faster convergence in (19).

[10]A visual illustration of this ellipse can be found in Azizian et al. (2020b, Figure 2).

[11]As GDM updates $w_{t+1}$ using both $w_t$ and $w_{t-1}$, Azizian et al. (2020b) uses an augmented fixed point operator; see Lemma 2 in that work for details.

**Theorem 8** (Azizian et al. (2020b)). *Define $\epsilon(\mu, L)$ as $\epsilon(\mu, L)/L = (\mu/L)^\theta = \tau^\theta$ with $\theta > 0$ and $a \wedge b = \min(a, b)$. If $Sp(\nabla F^{GDM}(w^\star, w^\star)) \subset K_\epsilon$, and when $\tau \to 0$, it satisfies:*

$$\rho(\nabla F^{GDM}(w^\star, w^\star)) \leqslant \begin{cases} 1 - 2\sqrt{\tau} + O\left(\tau^{\theta \wedge 1}\right), & \text{if } \theta > \frac{1}{2} \\ 1 - 2(\sqrt{2} - 1)\sqrt{\tau} + O(\tau), & \text{if } \theta = \frac{1}{2} \\ 1 - \tau^{1-\theta} + O\left(\tau^{1 \wedge (2-3\theta)}\right), & \text{if } \theta < \frac{1}{2}, \end{cases} \tag{30}$$

*where the hyperparametes $h$ and $m$ are functions of $\mu, L$, and $\epsilon$ only.*

For Case 1, GDM converges at the rate $1 - 2\sqrt{\tau} + O(\tau)$ (i.e., with $\theta = 1$ from the above), which is always slower than the rate of MEG in (18) by the factor of $\sqrt{1 - R^2}$. For Case 2, we see from Theorem 8 that GDM achieves an accelerated rate, i.e., $1 - O(\sqrt{\tau})$, until $\theta = \frac{1}{2}$. In other words, the biggest elliptic perturbation $\epsilon$ where GDM permits the accelerated rate is $\epsilon = \sqrt{\mu L}$.[12] We interpret Theorem 8 for games with cross-shaped Jacobian spectrum in (14) and shifted imaginary spectrum in (15) in the following corollary.

**Corollary 2.** *Consider the gradient method with momentum, interpreted as fixed point iteration as in (29). For games with cross-shaped Jacobian spectrum in (14) with $c = \frac{L-\mu}{2}$, GDM cannot achieve an accelerated rate when $\frac{L-\mu}{2} = c > \epsilon = \sqrt{\mu L}$. Since $L > \mu$, this further implies $\frac{L}{\mu} > \sqrt{5}$. That is, when the condition number exceeds $\sqrt{5} \approx 2.236$, GDM cannot achieve an accelerated convergence rate. On the contrary, as we showed in (19), MEG can converge at an accelerated rate in the ill-conditioned regime.*

The convergence rate of GDM for Case 3 cannot be determined from Theorem 8, as this theorem assumes the spectrum model of the real line segment $[\mu, L]$ with $\epsilon$ perturbation (along the imaginary axis), while $\mathcal{S}_3^\star$ in (15) has a fixed real component. Instead, we utilize the link function of GDM in (7) to show that it is unlikely for GDM to stay in the robust region: $\xi^{-1}([-1, 1])$.

**Proposition 2.** *Consider solving (1) for games where the Jacobian has a shifted imaginary spectrum in (15), using the gradient method with momentum in (5). For any complex number $z = p + qi \in \mathbb{C}_+$, if $\frac{2(1+m)}{h} < p$, then GDM cannot stay in the robust region, i.e., $|\xi(\lambda)| > 1$.*

Note that the condition $\frac{2(1+m)}{h} < p$ is hard to avoid even for small $p$, considering $h$ is usually a small value.

# 6 Local Convergence for Non-affine Vector Fields

The optimal hyperparameters of MEG for each spectrum model and the associated convergence rate we obtained in Section 4 are attainable when the vector field is affine. A natural question is, then, what can we say about the convergence rate of MEG when the vector field is not affine? To that end, we provide the local convergence of MEG by restarting the momentum, as detailed below.

Let us consider the operator $G$ representing the MEG in (4) such that:

$$[w_{t+1}, w_t] = G([w_t, w_{t-1}]) \quad \text{and} \quad G([w^\star, w^\star]) = [w^\star, w^\star].$$

In addition, we assume that $w_1 = w_0 - \frac{h}{1+m} v(w_0 - \gamma v(w_0))$, in order to induce the residual polynomials from Theorem 1; see also its proof and Algorithm 1 in the appendix. Now let us consider the following algorithm:

$$[w_{tk+i+1}, w_{tk+i}] = G([w_{tk+i}, w_{tk+i-1}]) \quad \text{for} \quad 1 \leqslant i \leqslant k - 1, \quad \text{and then}$$
$$w_{(t+1)k+1} = w_{(t+1)k} - \frac{h}{1+m} v(w_{(t+1)k} - \gamma v(w_{(t+1)k})). \tag{31}$$

In other words, we repeat MEG for $k$ steps, and then restart the momentum at $[w_{(t+1)k+1}, w_{(t+1)k}]$. The local convergence of the restarted MEG is established in the next theorem.

**Theorem 9** (Local convergence). *Let $G : \mathbb{R}^{2d} \to \mathbb{R}^{2d}$ be the continuously differentiable operator representing the momentum extragradient method (MEG) in (4). Let $w^\star$ be a stationary point. Let $w_t$ denote the output of MEG, which enjoys a convergence rate of the form $\|w_t - w^\star\| = C(1 - \varphi)^t (t + 1)\|w_0 - w^\star\|$ for some*

---

[12]Observe that $(\mu/L)^{1/2} = \epsilon(\mu, L)/L \implies \epsilon(\mu, L) = \sqrt{\mu L}$.

$0 < \varphi < 1$ *when the vector field is affine. Further, consider restarting the momentum of MEG after running* $k$ *steps, as in* (31)*. Then, for each* $\varepsilon > 0$*, there exists* $k > 0$ *and* $\delta > 0$ *such that, for all initializations* $w_0$ *satisfying* $\|w_0 - w^\star\| \leqslant \delta$*, the restarted MEG satisfies:*

$$\|w_t - w^\star\| = O((1 - \varphi + \varepsilon)^t)\|w_0 - w^\star\|.$$

## 7 Experiments

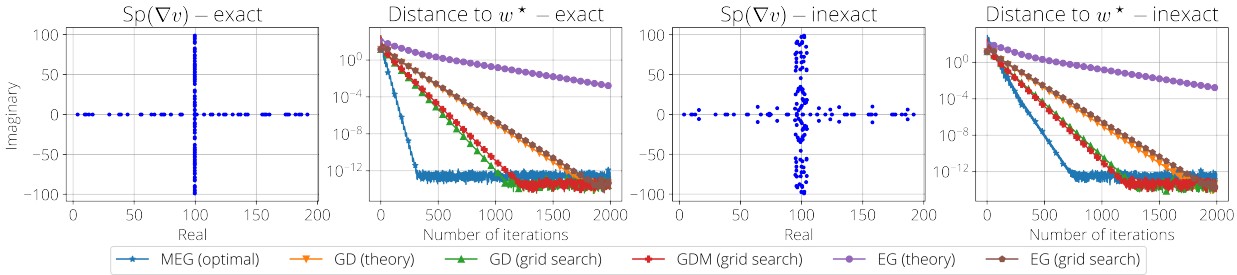

Figure 3: *Illustration of the game Jacobian spectra and the performance of different algorithms considered.* Jacobian spectrum in the first plot matches $\mathcal{S}_2^\star$ in (14) precisely, while that in the third plot inexactly follows $\mathcal{S}_2^\star$. The second (fourth) plot shows the performance of different algorithms for solving quadratic games in (16) with the Jacobian spectrum following the first (third) plot.

In this section, we perform numerical experiments to optimize a game when the Jacobian has a cross-shaped spectrum in (14). We focus on this spectrum as it may be the most challenging case, involving both real and complex eigenvalues (c.f., Theorem 3). To test the robustness, we consider two cases where the Jacobian spectrum exactly follows $\mathcal{S}_2^\star$ in (14), as well as the inexact case. We illustrate them in Figure 3.

We focus on two-player quadratic games, where player 1 controls $x \in \mathbb{R}^{d_1}$ and player 2 controls $y \in \mathbb{R}^{d_2}$ with loss functions in (16). In our setting, the corresponding vector field in (17) satisfies $M_{12} = -M_{21}^\top$, but $S_1$ and $S_2$ can be nonzero symmetric matrices. Further, the Jacobian $\nabla v = A$ has the cross-shaped eigenvalue structure in (14), with $c = \frac{L-\mu}{2}$ (c.f., Proposition 1, Case 2). For the problem constants, we use $\mu = 1$, and $L = 200$. The optimum $[x^\star\ y^\star]^\top = w^\star \in \mathbb{R}^{200}$ is generated using the standard normal distribution. For simplicity, we assume $b = [b_1\ b_2]^\top = [0\ 0]^\top$. For the algorithms, we compare GD in (22), GDM in (5), EG in (23), and MEG in (4). All algorithms are initialized with 0. We plot the experimental results in Figure 3.

For MEG (optimal), we set the hyperparameters using Theorem 5. For GD (theory) and EG (theory), we set the hyperparameters using Theorem 7, both for the exact and the inexact settings. For GDM (grid search), we perform a grid search of $h^{\text{GDM}}$ and $m^{\text{GDM}}$, and choose the best-performing ones, as Theorem 8 does not give a specific form for hyperparameter setup. Specifically, we consider $0.005 \leqslant h^{\text{GDM}} \leqslant 0.015$ with $10^{-3}$ increment, and $0.01 \leqslant m^{\text{GDM}} \leqslant 0.99$ with $10^{-2}$ increment. In addition, as Theorem 7 might be conservative, we conduct grid searches for GD and EG as well. For GD (grid search), we use the same setup as $h^{\text{GDM}}$. For EG (grid search), we use $0.001 \leqslant h^{\text{EG}} \leqslant 0.05$ with $10^{-4}$ increment.

There are several remarks to make. First, although the third plot in Figure 3 does not exactly follow the spectrum model in (14), MEG still works well with the optimal hyperparameters from Theorem 5. As expected, MEG (optimal) required more iterations in the inexact case compared to the exact case. Second, compared to other algorithms, MEG (optimal) indeed exhibits a significantly faster rate of convergence, even when compared to other methods that use grid-search hyperparameter tuning, supporting our theoretical findings in Section 4. Third, while EG (theory) is slower than GD (theory), which confirms Corrolary 1, EG (grid search) can be tuned to converge faster via grid search. Lastly, even though the best performance of GDM (grid search) is obtained through grid search, one can see the GD (grid search) obtains a slightly faster convergence rate than GDM (grid search), confirming Corollay 2.

## 8   Conclusion

In the study of differentiable games, finding stationary points efficiently is crucial. This work analyzes the momentum extragradient method, revealing three distinct convergence modes dependent on the Jacobian eigenvalue distribution. Through a polynomial-based analysis, we derive optimal hyperparameters for each mode, achieving accelerated asymptotic convergence rates. We compared the obtained rates with other first-order methods and showed that the considered methods do not achieve the accelerated convergence rate. Notably, our initial analysis for affine vector fields extends to guarantee local convergence rates on twice-differentiable vector fields. Numerical experiments on quadratic games validate our theoretical findings.

**Acknowledgments**

The authors would like to thank Fangshuo Liao, Baptiste Goujaud, Damien Scieur, Miri Son, and Giorgio Young for their useful discussions and feedback.

This work is supported by NSF FET: Small No. 1907936, NSF MLWiNS CNS No. 2003137 (in collaboration with Intel), NSF CMMI No. 2037545, NSF CAREER award No. 2145629, NSF CIF No. 2008555, Rice InterDisciplinary Excellence Award (IDEA), and the Canada CIFAR AI Chairs program.

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

# A  Missing Proofs in Section 3

## A.1  Proof of Lemma 1

Proof of Lemma 1 can be found for example in Azizian et al. (2020b, Section B).

## A.2  Proof of Theorem 1

To obtain the residual polynomials of MEG, $w_1$ has to be set slightly differently from the rest of the iterates, as we write in the pseudocode below:

---

**Algorithm 1:** Momentum extragradient (MEG) method

---

**Input:** Initialization $w_0$, hyperparameters $h, \gamma, m$.
**Set:** $w_1 = w_0 - \frac{h}{1+m} v(w_0 - \gamma v(w_0))$
**for** $t = 1, 2, \ldots$ **do**
  | $w_{t+1} = w_t - hv(w_t - \gamma v(w_t)) + m(w_t - w_{t-1})$
**end**

---

**Derivation of the first part**

*Proof.* We want to find the residual polynomial $\tilde{P}_t(A)$ of the extragradient with momentum (MEG) in (4). That is, we want to find

$$w_t - w^\star = \tilde{P}_t(A)(w_0 - w^\star), \tag{32}$$

where $\{w_t\}_{t=0}$ is the iterates generated by MEG, which is possible by Lemma 1, as MEG is a first-order method (Arjevani & Shamir, 2016; Azizian et al., 2020b). We now prove this is by induction. To do so, we will use the following properties. First, note that as we are looking for a stationary point, it holds that $v(w^\star) = 0$. Further, as $v$ is linear by the assumption of Lemma 1, it holds that $v(w) = A(w - w^\star)$.

**Base case.** For $t = 0$, $\tilde{P}_0(A)$ is a degree-zero polynomial, and hence equals $I_d$, which denotes the identity matrix. Thus, $w_0 - w^\star = I_d(w_0 - w^\star)$ holds true.

For completeness, we also prove when $t = 1$. In that case, observe that MEG in proceeds as $w_1 = w_0 - \frac{h}{1+m} v(w_0 - \gamma v(w_0))$. Subtracting $w^\star$ on both sides, we have:

$$
\begin{aligned}
w_1 - w^\star &= w_0 - w^\star - \frac{h}{1+m} v(w_0 - \gamma v(w_0)) \\
&= w_0 - w^\star - \frac{h}{1+m} v(w_0 - \gamma A(w_0 - w^\star)) \\
&= w_0 - w^\star - \frac{h}{1+m} A(w_0 - \gamma A(w_0 - w^\star) - w^\star) \\
&= w_0 - w^\star - \frac{h}{1+m} A(w_0 - w^\star) + \frac{h\gamma}{1+m} A^2(w_0 - w^\star) \\
&= \left( I_d - \frac{h}{1+m} A + \frac{h\gamma}{1+m} A^2 \right)(w_0 - w^\star) \\
&= \left( I_d - \frac{h}{1+m} A(I_d - \gamma A) \right)(w_0 - w^\star) \\
&= \tilde{P}_1(A)(w_0 - w^\star).
\end{aligned}
$$

**Induction step.** As the induction hypothesis, assume $\tilde{P}_t$ satisfies (32). We want to prove this holds for $t + 1$. We have:

$$
\begin{aligned}
w_{t+1} &= w_t - hv(w_t - \gamma v(w_t)) + m(w_t - w_{t-1}) \\
&= w_t - hv(w_t - \gamma A(w_t - w^\star)) + m(w_t - w_{t-1}) \\
&= w_t - hA(w_t - \gamma A(w_t - w^\star) - w^\star) + m(w_t - w_{t-1}) \\
&= w_t - hA(w_t - w^\star) + h\gamma A^2(w_t - w^\star) + m(w_t - w_{t-1}) \\
&= w_t - hA(I_d - \gamma A)(w_t - w^\star) + m(w_t - w_{t-1}).
\end{aligned}
$$

Subtracting $w^\star$ on both sides, we have:

$$
\begin{aligned}
w_{t+1} - w^\star &= w_t - w^\star - hA(I_d - \gamma A)(w_t - w^\star) + m(w_t - w_{t-1}) \\
&= (I_d - hA(I_d - \gamma A))(w_t - w^\star) + m(w_t - w^\star - (w_{t-1} - w^\star)) \\
&\stackrel{(32)}{=} (I_d - hA(I_d - \gamma A))\tilde{P}_t(A)(w_0 - w^\star) + m(\tilde{P}_t(A)(w_0 - w^\star) - \tilde{P}_{t-1}(A)(w_0 - w^\star)) \\
&= (I_d + mI_d - hA(I_d - \gamma A))\tilde{P}_t(A)(w_0 - w^\star) - m\tilde{P}_{t-1}(A)(w_0 - w^\star) \\
&= \tilde{P}_{t+1}(A)(w_0 - w^\star),
\end{aligned}
$$

where in the third equality, we used the induction hypothesis in (32).

$\square$

**Derivation of the second part in** (6)

*Proof.* We show $P_t = \tilde{P}_t$ for all $t$ via induction.

**Base case.** For $t = 0$, by the definition of Chebyshev polynomials of the first and the second kinds, we have $T_0(\lambda) = U_0(\lambda) = 1$. Thus,

$$
\begin{aligned}
P_0(\lambda) &= m^0 \left( \frac{2m}{1+m} T_0(\sigma(\lambda)) + \frac{1-m}{1+m} U_0(\sigma(\lambda)) \right) \\
&= \frac{2m}{1+m} + \frac{1-m}{1+m} = 1 = \tilde{P}_0(\lambda).
\end{aligned}
$$

Again, for completeness, we prove when $t = 1$ as well. In that case, by the definition of Chebyshev polynomials of the first and the second kinds, we have $T_1(\lambda) = \lambda$, and $U_1(\lambda) = 2\lambda$. Therefore,

$$
\begin{aligned}
P_1(\lambda) &= m^{t/2} \left( \frac{2m}{1+m} T_1(\sigma(\lambda)) + \frac{1-m}{1+m} U_1(\sigma(\lambda)) \right) \\
&= m^{t/2} \left( \frac{2m}{1+m} \sigma(\lambda) + \frac{1-m}{1+m} \cdot 2 \cdot \sigma(\lambda) \right) \\
&= m^{t/2} \left( \frac{2\sigma(\lambda)}{1+m} \right) \\
&= 1 - \frac{h\lambda(1 - \gamma\lambda)}{1+m} = \tilde{P}_1(\lambda).
\end{aligned}
$$

**Induction step.** As the induction hypothesis, assume that $P_t = \tilde{P}_t$ for $t$. In this step, we show that the same holds for $t + 1$.

$$
\begin{aligned}
P_{t+1} &= m^{(t+1)/2}\left[\frac{2m}{1+m}T_{t+1}(\sigma(\lambda)) + \frac{1-m}{1+m}U_{t+1}(\sigma(\lambda))\right] \\
&= m^{(t+1)/2}\left[\frac{2m}{1+m}\Big(2\sigma(\lambda)T_t(\sigma(\lambda)) - T_{t-1}(\sigma(\lambda))\Big)\right. \\
&\qquad\qquad \left.+ \frac{1-m}{1+m}\Big(2\sigma(\lambda)U_t(\sigma(\lambda) - U_{t-1}(\sigma(\lambda)))\Big)\right] \\
&= 2\sigma(\lambda)\cdot m^{1/2}\cdot \underbrace{m^{t/2}\left(\frac{2m}{1+m}T_t(\sigma(\lambda)) + \frac{1-m}{1+m}U_t(\sigma(\lambda))\right)}_{P_t(\lambda)} \\
&\qquad - m\cdot \underbrace{m^{(t-1)/2}\left(\frac{2m}{1+m}T_{t-1}(\sigma(\lambda)) + \frac{1-m}{1+m}U_{t-1}(\sigma(\lambda))\right)}_{P_{t-1}(\lambda)} \\
&= 2\sigma(\lambda)\cdot \sqrt{m}\cdot \tilde{P}_t(\lambda) - m\cdot \tilde{P}_{t-1}(\lambda) \\
&= (1 + m - h\lambda(1 - \gamma\lambda))\tilde{P}_t(\lambda) - m\tilde{P}_{t-1}(\lambda),
\end{aligned}
$$

where in the second to last equality we use the induction hypothesis. $\qquad\square$

### A.3 Proof of Lemma 2

Proof of Lemma 2 can be found in Goujaud & Pedregosa (2022).

### A.4 Proof of Theorem 2

*Proof.* We first recall that using (3), we can upper bound the worst-case convergence rate as:

$$
\begin{aligned}
\sup_{\lambda\in\mathcal{S}^\star}|P_t(\lambda)| &\overset{(6)}{=} \sup_{\lambda\in\mathcal{S}^\star}\left|m^{t/2}\left(\tfrac{2m}{1+m}T_t(\sigma(\lambda)) + \tfrac{1-m}{1+m}U_t(\sigma(\lambda))\right)\right| \\
&\leqslant m^{t/2}\left(\tfrac{2m}{1+m}\sup_{\lambda\in\mathcal{S}^\star}|T_t(\sigma(\lambda))| + \tfrac{1-m}{1+m}\sup_{\lambda\in\mathcal{S}^\star}|U_t(\sigma(\lambda))|\right)
\end{aligned}
\tag{33}
$$

Now, denote $\bar{\sigma} := \sup_{\lambda\in\mathcal{S}^\star}|\sigma(\lambda; h, \gamma, m)|$. For the first case, if $\bar{\sigma}\leqslant 1$, both $T_t(x)$ and $U_t(x)$ behave nicely, per Lemma 2. Thus, we have

$$
(33)\overset{(8)}{\leqslant} m^{t/2}\left(\frac{2m}{1+m} + \frac{1-m}{1+m}(t+1)\right) \leqslant m^{t/2}(t+1) \implies \limsup_{t\to\infty}\left(m^{t/2}(t+1)\right)^{\frac{1}{2t}} = \sqrt[4]{m}.
$$

For the second case, we use the following expressions of Chebyshev polynomials:

$$
T_n(x) = \frac{\left(x - \sqrt{x^2 - 1}\right)^n + \left(x + \sqrt{x^2 - 1}\right)^n}{2}, \quad \text{and}
$$

$$
U_n(x) = \frac{\left(x + \sqrt{x^2 - 1}\right)^{n+1} - \left(x - \sqrt{x^2 - 1}\right)^{n+1}}{2\sqrt{x^2 - 1}}.
$$

Therefore, in the second case where $\bar{\sigma} > 1$, we have both $T_n(x)$ and $U_n(x)$ growing at rate $(x + \sqrt{x^2 - 1})^n$. Hence, we have:

$$
(33)\leqslant O\left(m^{t/2}\left(\bar{\sigma} + \sqrt{\bar{\sigma}^2 - 1}\right)^t\right) \implies \limsup_{t\to\infty}\left(m^{t/2}\left(\bar{\sigma} + \sqrt{\bar{\sigma}^2 - 1}\right)^t\right)^{\frac{1}{2t}} = \sqrt[4]{m}\left(\bar{\sigma} + \sqrt{\bar{\sigma}^2 - 1}\right)^{1/2}.
$$

Finally, in order for MEG to converge in the second case, we need:

$$\sqrt[4]{m}\left(\bar{\sigma}+\sqrt{\bar{\sigma}^2-1}\right)^{1/2} < 1$$

which is equivalent to

$$\bar{\sigma} \leqslant \frac{\sqrt{m}(m+1)}{2m} = \frac{m+1}{2\sqrt{m}}.$$

$\square$

## A.5 Derivation of extreme points of robust region in (11)

We first write a general formula for inverting a quadratic function. For $f(x) = ax^2 + bx + c$, its inverse is given by:

$$f(x) = ax^2 + bx + c := y$$
$$f^{-1}(y) = \frac{-b \pm \sqrt{b^2 - 4a(c-y)}}{2a},$$

with some abuse of notation (i.e., $f^{-1}$ above is not a function).

Applying the above to the link function of MEG in (6), we get

$$\sigma^{-1}(y) = \frac{1}{2\gamma} \pm \sqrt{\frac{1}{4\gamma^2} - \frac{1+m}{h\gamma} + \frac{2\sqrt{m}}{h\gamma} \cdot y}.$$

With this formula, we can plug in 1 and $-1$ to get:

$$\sigma^{-1}(-1) = \frac{1}{2\gamma} \pm \sqrt{\frac{1}{4\gamma^2} - \frac{(1+\sqrt{m})^2}{h\gamma}} \quad \text{and} \quad \sigma^{-1}(1) = \frac{1}{2\gamma} \pm \sqrt{\frac{1}{4\gamma^2} - \frac{(1-\sqrt{m})^2}{h\gamma}}.$$

## A.6 Proof of Theorem 3

*Proof.* We analyze each case separately.

**Case 1:** There are two square roots: $\sqrt{\frac{1}{4\gamma^2} - \frac{(1-\sqrt{m})^2}{h\gamma}}$ and $\sqrt{\frac{1}{4\gamma^2} - \frac{(1+\sqrt{m})^2}{h\gamma}}$. The second one is real if:

$$\frac{1}{4\gamma^2} \geqslant \frac{(1+\sqrt{m})^2}{h\gamma} \implies \frac{h\gamma}{4\gamma^2} = \frac{h}{4\gamma} \geqslant (1+\sqrt{m})^2,$$

which implies the first is real, as $(1+\sqrt{m})^2 \geqslant (1-\sqrt{m})^2$.

**Case 3:** There are two square roots: $\sqrt{\frac{1}{4\gamma^2} - \frac{(1-\sqrt{m})^2}{h\gamma}}$ and $\sqrt{\frac{1}{4\gamma^2} - \frac{(1+\sqrt{m})^2}{h\gamma}}$. The first one is complex if:

$$\frac{1}{4\gamma^2} < \frac{(1-\sqrt{m})^2}{h\gamma} \implies \frac{h\gamma}{4\gamma^2} = \frac{h}{4\gamma} < (1-\sqrt{m})^2,$$

which implies the second is complex, as $(1+\sqrt{m})^2 \geqslant (1-\sqrt{m})^2$.

**Case 2:** This case follows automatically from the above two cases.

$\square$

### A.7 Proof of Proposition 1

*Proof.* Define $D_3 = \text{diag}(a, \ldots, a)$ of dimensions $d \times d$. Let us prove that if there exists $U, V$ orthonormal matrices and $D_1, D_2$ matrices with non-zeros coefficients only on the diagonal such that (with a slight abuse of notation)

$$S_1 = U\text{diag}(D_3, D_1)U^\top, S_2 = VD_3V^\top, \quad \text{and} \quad B = UD_2V^\top,$$

then the spectrum of $A$ is crossed shaped. In that case, we have

$$A = \begin{bmatrix} U[D_3; D_1]U^\top & UD_2V^\top, \\ -VD_2^\top U^\top & VD_3V^\top \end{bmatrix}$$

$$= \begin{bmatrix} U & 0 \\ 0 & V \end{bmatrix} \begin{bmatrix} [D_3; D_1] & D_2 \\ -D_2^\top & D_3 \end{bmatrix} \begin{bmatrix} U & 0 \\ 0 & V \end{bmatrix}^\top.$$

Now by considering the basis $W = ((U_1, 0), (0, V_1), \ldots, (U_{d_v}, 0), (0, V_{d_v}), (U_{d_v+1}, 0), \ldots, (U_d, 0))$ we have that $A$ can be block diagonalized in that basis as

$$A = W\text{diag}\left(\begin{bmatrix} a & [D_2]_{11} \\ -[D_2]_{11} & a \end{bmatrix}, \ldots, \begin{bmatrix} a & [D_2]_{d_v, d_v} \\ -[D_2]_{d_v d_v} & a \end{bmatrix}, [D_1]_1, \ldots, [D_1]_{d_u-d_v d_u-d_v}\right) W^\top. \tag{34}$$

Now, notice that

$$\text{Sp}\left(\begin{bmatrix} a & -b \\ b & a \end{bmatrix}\right) = \{a \pm bi\}, \tag{35}$$

since the associated characteristic polynomial of the above matrix is:

$$(a-\lambda)^2 + b^2 = 0 \implies a - \lambda = \pm bi \implies \lambda = a \pm bi.$$

Hence, using (35) in the formulation of $A$ in (34), we have that the spectrum of $A$ is cross-shaped.

$\square$

## B Missing Proofs in Section 4

### B.1 Proof of Theorem 4

*Proof.* We write the conditions required for Theorem 5 below:

$$\frac{1}{2\gamma} - \sqrt{\frac{1}{4\gamma^2} - \frac{(1-\sqrt{m})^2}{h\gamma}} = \mu_1, \tag{36}$$

$$\frac{1}{2\gamma} - \sqrt{\frac{1}{4\gamma^2} - \frac{(1+\sqrt{m})^2}{h\gamma}} = L_1, \tag{37}$$

$$\frac{1}{2\gamma} + \sqrt{\frac{1}{4\gamma^2} - \frac{(1+\sqrt{m})^2}{h\gamma}} = \mu_2, \quad \text{and} \tag{38}$$

$$\frac{1}{2\gamma} + \sqrt{\frac{1}{4\gamma^2} - \frac{(1-\sqrt{m})^2}{h\gamma}} = \mu_2. \tag{39}$$

By adding (37) and (38) (or equivalently by ading (36) and (39)), we get

$$\gamma = \frac{1}{\mu_1 + L_2} = \frac{1}{\mu_2 + L_1}. \tag{40}$$

From (36), we have:

$$\frac{1}{2\gamma} + \sqrt{\frac{1}{4\gamma^2} - \frac{(1-\sqrt{m})^2}{h\gamma}} = \mu_1$$

$$\frac{1}{4\gamma^2} - \frac{(1-\sqrt{m})^2}{h\gamma} = \left(\frac{1}{2\gamma} - \mu_1\right)^2$$

$$\frac{(1-\sqrt{m})^2}{h} = \mu_1(1 - \gamma\mu_1)$$

$$h = \frac{(1-\sqrt{m})^2}{\mu_1(1 - \gamma\mu_1)} = \frac{(1-\sqrt{m})^2(\mu_1 + L_2)}{\mu_1 L_2} \tag{41}$$

Similarly, from (38), we have:

$$\frac{1}{2\gamma} + \sqrt{\frac{1}{4\gamma^2} - \frac{(1+\sqrt{m})^2}{h\gamma}} = \mu_2$$

$$\frac{1}{4\gamma^2} - \frac{(1+\sqrt{m})^2}{h\gamma} = \left(\frac{\mu_2 - L_1}{2}\right)^2$$

$$\left(\frac{\mu_2 + L_1}{2}\right)^2 - \left(\frac{\mu_2 - L_1}{2}\right)^2 = \mu_2 L_1 = \frac{(1+\sqrt{m})^2}{h\gamma} \tag{42}$$

Combining (41) and (42), and solving for $m$, we get

$$\mu_2 L_1(1 - \sqrt{m})^2 = \mu_1 L_2(1 + \sqrt{m})^2$$

$$m = \left(\frac{\sqrt{\mu_2 L_1} - \sqrt{\mu_1 L_2}}{\sqrt{\mu_2 L_1} + \sqrt{\mu_1 L_2}}\right)^2 \overset{(13)}{=} \left(\frac{\sqrt{\zeta^2 - R^2} - \sqrt{\zeta^2 - 1}}{\sqrt{\zeta^2 - R^2} + \sqrt{\zeta^2 - 1}}\right)^2. \tag{43}$$

Finally, plugging (43) back to (41), we get:

$$h = \frac{(1-\sqrt{m})^2(\mu_1 + L_2)}{\mu_1 L_2}$$

$$= \frac{4\mu_1 L_2}{(\sqrt{\mu_2 + L_1} + \sqrt{\mu_1 + L_2})^2} \cdot \frac{\mu_1 + L_2}{\mu_1 L_2}$$

$$= \frac{4(\mu_1 + L_2)}{(\sqrt{\mu_2 + L_1} + \sqrt{\mu_1 + L_2})^2}.$$

$\square$

## B.2  Proof of Theorem 5

*Proof.* We write the conditions required for Theorem 5 below:

$$\frac{1}{2\gamma} - \sqrt{\frac{1}{4\gamma^2} - \frac{(1-\sqrt{m})^2}{h\gamma}} = \mu, \tag{44}$$

$$\frac{1}{2\gamma} + \sqrt{\frac{1}{4\gamma^2} - \frac{(1-\sqrt{m})^2}{h\gamma}} = L, \quad \text{and} \tag{45}$$

$$\sqrt{\frac{(1+\sqrt{m})^2}{h\gamma} - \frac{1}{4\gamma^2}} = c. \tag{46}$$

First, by adding (44) and (45), we get:

$$\frac{1}{\gamma} = \mu + L \implies \gamma = \frac{1}{\mu + L}. \tag{47}$$

Plugging (47) back into (44), we have:

$$\frac{1}{2\gamma} - \sqrt{\frac{1}{4\gamma^2} - \frac{(1-\sqrt{m})^2}{h\gamma}} = \mu$$

$$\frac{\mu+L}{2} - \mu = \sqrt{\left(\frac{\mu+L}{2}\right)^2 - \frac{(1-\sqrt{m})^2(\mu+L)}{h}}$$

$$\left(\frac{L-\mu}{2}\right)^2 = \left(\frac{\mu+L}{2}\right)^2 - \frac{(1-\sqrt{m})^2(\mu+L)}{h}$$

$$\frac{(1-\sqrt{m})^2(\mu+L)}{h} = \left(\frac{\mu+L}{2}\right)^2 - \left(\frac{L-\mu}{2}\right)^2 = \mu L$$

$$h = \frac{(1-\sqrt{m})^2(\mu+L)}{\mu L}. \tag{48}$$

Plugging (47) and (48) into (46), we have:

$$\sqrt{\frac{(1+\sqrt{m})^2}{h\gamma} - \frac{1}{4\gamma^2}} = c$$

$$\sqrt{\frac{(1+\sqrt{m})^2 \cdot \mu L}{(1-\sqrt{m})^2} - \left(\frac{\mu+L}{2}\right)^2} = c$$

$$\frac{(1+\sqrt{m})^2 \cdot \mu L}{(1-\sqrt{m})^2} = c^2 + \left(\frac{\mu+L}{2}\right)^2 = \frac{4c^2 + (\mu+L)^2}{4}$$

$$\frac{(1+\sqrt{m})^2}{(1-\sqrt{m})^2} = \frac{4c^2 + (\mu+L)^2}{4\mu L}$$

$$(1+\sqrt{m})\sqrt{4\mu L} = (1-\sqrt{m})\sqrt{4c^2 + (\mu+L)^2}$$

$$\sqrt{m}(\sqrt{4c^2 + (\mu+L)^2} + \sqrt{4\mu L}) = \sqrt{4c^2 + (\mu+L)^2} - \sqrt{4\mu L}$$

$$\sqrt{m} = \frac{\sqrt{4c^2 + (\mu+L)^2} - \sqrt{4\mu L}}{\sqrt{4c^2 + (\mu+L)^2} + \sqrt{4\mu L}}. \tag{49}$$

Finally, to simplify (48) further, from (49), we have:

$$1 - \sqrt{m} = \frac{4\sqrt{\mu L}}{\sqrt{4c^2 + (\mu+L)^2} + \sqrt{4\mu L}}.$$

Hence, from (48),

$$h = \frac{(\mu+L)(1-\sqrt{m})^2}{\mu L} = \frac{\frac{16\mu L(\mu+L)}{(\sqrt{4c^2+(\mu+L)^2}+\sqrt{4\mu L})^2}}{\mu L} = \frac{16(\mu+L)}{(\sqrt{4c^2+(\mu+L)^2}+\sqrt{4\mu L})^2}. \tag{50}$$

$$\square$$

### B.3 Proof of Theorem 6

*Proof.* We write the conditions required for (6) below:

$$\frac{1}{2\gamma} + \sqrt{\frac{1}{4\gamma^2} - \frac{(1+\sqrt{m})^2}{h\gamma}} = c + bi, \tag{51}$$

$$\frac{1}{2\gamma} - \sqrt{\frac{1}{4\gamma^2} - \frac{(1+\sqrt{m})^2}{h\gamma}} = c + ai, \tag{52}$$

$$\frac{1}{2\gamma} + \sqrt{\frac{1}{4\gamma^2} - \frac{(1+\sqrt{m})^2}{h\gamma}} = c - ai, \quad \text{and} \tag{53}$$

$$\frac{1}{2\gamma} + \sqrt{\frac{1}{4\gamma^2} - \frac{(1-\sqrt{m})^2}{h\gamma}} = c - bi. \tag{54}$$

First, we can see from all cases that the optimal $\gamma$ is

$$\gamma = \frac{1}{2c}. \tag{55}$$

(51) and (54) equivalently imply

$$\sqrt{\frac{(1+\sqrt{m})^2}{h\gamma} - \frac{1}{4\gamma^2}} = b$$

$$(1+\sqrt{m})^2 = h\gamma b^2 + \frac{h}{4\gamma} = \frac{h(c^2+b^2)}{2c}$$

$$h = \frac{2c(1+\sqrt{m})^2}{c^2+b^2}. \tag{56}$$

Similarly, (52) and (53) imply

$$\sqrt{\frac{(1-\sqrt{m})^2}{h\gamma} - \frac{1}{4\gamma^2}} = a$$

$$\frac{(1-\sqrt{m})^2}{h\gamma} = a^2 + \frac{1}{4\gamma^2} = a^2 + c^2$$

$$\frac{(1-\sqrt{m})^2(c^2+b^2)}{(1+\sqrt{m})^2} = a^2 + c^2$$

$$(1-\sqrt{m})\sqrt{c^2+b^2} = (1+\sqrt{m})\sqrt{c^2+a^2}$$

$$\sqrt{m} = \frac{\sqrt{c^2+b^2} - \sqrt{c^2+a^2}}{\sqrt{c^2+b^2} + \sqrt{c^2+a^2}} = 1 - \frac{2\sqrt{c^2+a^2}}{\sqrt{c^2+b^2} + \sqrt{c^2+a^2}} \tag{57}$$

Plugging (57) to (56), we get

$$h = \frac{2c(1+\sqrt{m})^2}{c^2+b^2} = \frac{8c}{(\sqrt{c^2+b^2} + \sqrt{c^2+a^2})^2}.$$

$\square$

## C Missing Proofs in Sections 5

### C.1 Proof of Corollary 1

*Proof.* To compute the convergence rates of GD and EG from Theorem 7 applied to each spectrum model in (12), (14), and (15), we need to compute $\min_{\lambda \in \Delta^\star} \mathfrak{R}(1/\lambda)$ and $\min_{\lambda \in \Delta^\star} \mathfrak{R}(\lambda)$ for GD. Similarly for EG, we need to compute additionally $\min_{\lambda \in \Delta^\star} |\lambda|$, $\min_{\lambda \in \Delta^\star} |\lambda|^2$, and $\sup_{\lambda \in \Delta^\star} |\lambda|^2$.

**Case 1:** It's straightforward to compute

$$\min_{\lambda \in \mathcal{S}_1^\star} \mathfrak{R}(1/\lambda) = 1/L_2, \quad \text{and} \quad \min_{\lambda \in \mathcal{S}_1^\star} \mathfrak{R}(\lambda) = \mu_1$$

Thus, GD for Case 1 has the rate

$$1 - \frac{\mu_1}{L_2} = 1 - \tau.$$

For EG, it's also simple to obtain

$$\min_{\lambda \in \mathcal{S}_1^\star} |\lambda| = L_2, \quad \min_{\lambda \in \mathcal{S}_1^\star} |\lambda|^2 = \mu_1^2, \quad \text{and} \quad \sup_{\lambda \in \mathcal{S}_1^\star} |\lambda|^2 = L_2^2.$$

Thus, EG for Case 1 has the rate

$$1 - \frac{1}{4}\left(\frac{\mu_1}{L_2} + \frac{1}{16}\left(\frac{\mu_1}{L_2}\right)^2\right).$$

**Case 2:** For a complex number $z = p + qi \in \mathbb{C}$, we can compute $\mathfrak{R}(1/z)$ as:

$$\frac{1}{z} = \frac{1}{p + qi} = \frac{p - qi}{p^2 + q^2} = \frac{p}{p^2 + q^2} - \frac{q}{p^2 + q^2}i \implies \mathfrak{R}\left(\frac{1}{z}\right) = \frac{p}{p^2 + q^2}.$$

The four extreme points of the cross-shaped spectrum model in (14) are:

$$\mu = \mu + 0i, \quad L = L + 0i, \quad \text{and} \quad \frac{L - \mu}{2} \pm ci.$$

Hence, $\mathfrak{R}(1/z)$ for each of the above points is:

$$\mathfrak{R}\left(\frac{1}{\mu}\right) = \frac{\mu}{\mu^2} = \frac{1}{\mu},$$

$$\mathfrak{R}\left(\frac{1}{L}\right) = \frac{L}{L^2} = \frac{1}{L}, \quad \text{and}$$

$$\mathfrak{R}\left(\frac{1}{\frac{L-\mu}{2} \pm ci}\right) = \frac{\frac{L-\mu}{2}}{\left(\frac{L-\mu}{2}\right)^2 + c^2}$$

$$= \frac{2(L - \mu)}{4c^2 + (L - \mu)^2}.$$

Therefore, $\min_{\lambda \in \mathcal{S}_2^\star} \mathfrak{R}\left(\frac{1}{\lambda}\right) = \frac{1}{L}$. As $\mu < L$, we only need to compare the last two values. Observe that:

$$c > \sqrt{\frac{L^2 - \mu^2}{4}}$$
$$4c^2 > (L - \mu)(L + \mu)$$
$$4c^2 > 2L(L - \mu) - (L - \mu)^2$$
$$\frac{1}{L} > \frac{2(L - \mu)}{4c^2 + (L - \mu)^2}.$$

Therefore,

$$\min_{\lambda \in \mathcal{S}_2^\star} \mathfrak{R}\left(\frac{1}{\lambda}\right) = \begin{cases} \frac{2(L-\mu)}{4c^2 + (L-\mu)^2} & \text{if} \quad c > \sqrt{\frac{L^2 - \mu^2}{4}} \\ \frac{1}{L} & \text{otherwise.} \end{cases}$$

For $\min_{\lambda \in \mathcal{S}_2^\star} \mathfrak{R}(\lambda)$, it's straightforward from the definition that

$$\min_{\lambda \in \mathcal{S}_2^\star} \mathfrak{R}(\lambda) = \mu.$$

Thus, GD for Case 2 has the rate

$$\begin{cases} 1 - \frac{2\mu(L-\mu)}{4c^2 + (L-\mu)^2} & \text{if} \quad c > \sqrt{\frac{L^2 - \mu^2}{4}} \\ 1 - \frac{\mu}{L} & \text{otherwise.} \end{cases}$$

Similarly for EG, we need to compute $\min_{\lambda \in \mathcal{S}_2^\star} \mathfrak{R}(\lambda)$, which was computed above; additionally, we need to compute $\min_{\lambda \in \mathcal{S}_2^\star} |\lambda|$, $\min_{\lambda \in \mathcal{S}_2^\star} |\lambda|^2$, and $\sup_{\lambda \in \mathcal{S}_2^\star} |\lambda|^2$. For $z = p + qi \in \mathbb{C}$, $|z| = \sqrt{p^2 + q^2}$. Hence, we have

$$|\mu + 0i| = \mu, \quad |L + 0i| = L, \quad \text{and} \quad \left| \frac{L-\mu}{2} \pm ci \right| = \sqrt{c^2 + \left( \frac{L-\mu}{2} \right)^2}.$$

Observe that:

$$c > \sqrt{\frac{3L^2 + 2L\mu - \mu^2}{4}}$$

$$c^2 > L^2 - \frac{L^2 - 2L\mu + \mu^2}{4}$$

$$c^2 + \left( \frac{L-\mu}{2} \right)^2 > L^2$$

Thus, for $\sup_{\lambda \in \mathcal{S}_2^\star} |\lambda|$, we have

$$\sup_{\lambda \in \mathcal{S}_2^\star} |\lambda| = \begin{cases} \sqrt{c^2 + \left( \frac{L-\mu}{2} \right)^2} & \text{if} \quad c > \sqrt{\frac{3L^2 + 2L\mu - \mu^2}{4}} \\ L & \text{otherwise,} \end{cases}$$

from which $\sup_{\lambda \in \mathcal{S}_2^\star} |\lambda|^2$ can also be obtained. Lastly, $\min_{\lambda \in \mathcal{S}_2^\star} |\lambda|^2 = \mu^2$, as we know $\mu < L$, and $(L-\mu)/2$ is the center of $[\mu, L]$.

Combining all three, we get the rate of EG for Case 3 is

$$\begin{cases} 1 - \frac{1}{4} \left( \frac{\mu}{\sqrt{c^2 + \left( \frac{L-\mu}{2} \right)^2}} + \frac{\mu^2}{16(c^2 + (\frac{L-\mu}{2})^2)} \right) & \text{if } c \geqslant \sqrt{\frac{3L^2 + 2L\mu - \mu^2}{4}}, \\ 1 - \frac{1}{4} \left( \frac{\mu}{L} + \frac{\mu^2}{16L^2} \right) & \text{otherwise.} \end{cases}$$

**Case 3:** Since (15) has fixed real component, $\min_{\lambda \in \mathcal{S}_3^\star} \mathfrak{R}(\lambda) = c$.

For $\min_{\lambda \in \mathcal{S}_3^\star} \mathfrak{R}(1/\lambda)$ we can compute compare

$$\mathfrak{R} \left( \frac{1}{c + ai} \right) = \frac{c}{c^2 + a^2} > \frac{c}{c^2 + b^2} = \mathfrak{R} \left( \frac{1}{c + bi} \right),$$

since $a < b$ from (15). Thus, GD for Case 3 has the rate

$$1 - \frac{c^2}{c^2 + b^2}.$$

For EG, it's also simple to obtain

$$\min_{\lambda \in \mathcal{S}_3^\star} |\lambda| = \sqrt{c^2 + b^2}, \quad \min_{\lambda \in \mathcal{S}_3^\star} |\lambda|^2 = c^2 + a^2, \quad \text{and} \quad \sup_{\lambda \in \mathcal{S}_3^\star} |\lambda|^2 = c^2 + b^2.$$

Thus, EG has the rate

$$1 - \frac{1}{4} \left( \frac{c}{\sqrt{c^2 + b^2}} + \frac{1}{16} \frac{(c^2 + a^2)}{(c^2 + b^2)} \right).$$

$\square$

## C.2   Proof of Corollary 2

*Proof.* Per Theorem 8, the largest $\epsilon$ that permits acceleration for GDM is $\epsilon = \sqrt{\mu L}$. Therefore, in the special case of (14) we consider, i.e., when $c = \frac{L-\mu}{2}$, GDM *cannot* achieve acceleration if $\frac{L-\mu}{2} > \sqrt{\mu L}$. Hence, we have:

$$\frac{L-\mu}{2} > \sqrt{\mu L}$$
$$L - \mu > 2\sqrt{\mu L}$$
$$L^2 + \mu^2 > 6\mu L > 6\mu^2 \quad (\because L > \mu)$$
$$L > \sqrt{5}\mu.$$

$\square$

## C.3   Proof of Proposition 2

*Proof.* For an arbitrary complex number $p + qi$ with $p > 0$, and using the link function of GDM from (7), we have

$$|\xi(p+qi)| = \sqrt{\left(\frac{1+m-hp}{2\sqrt{m}}\right)^2 + \left(\frac{hq}{2\sqrt{m}}\right)^2} \leqslant 1$$
$$\frac{(1+m-hp)^2 + h^2q^2}{4m} \leqslant 1$$
$$(1+m-hp)^2 + h^2q^2 \leqslant 4m$$
$$(1-m)^2 + hp(hp - 2(1+m)) + h^2q^2 \leqslant 0$$
$$\frac{(1-m)^2 + h^2q^2}{hp} \leqslant 2(1+m) - hp$$

Notice that the LHS is positive. Therefore, if the RHS is negative, the above inequality cannot hold. In other words, if $\frac{2(1+m)}{h} < p$, GDM cannot stay in the robust region. This is very hard to satisfy, even with a small $p$.

$\square$

## D   Missing Proofs in Section 6

Let us consider an affine vector field $v(w) = Aw + b$ and its associated augmented MEG linear operator $J$:

$$\begin{bmatrix} w_{t+1} - w^\star \\ w_t - w^\star \end{bmatrix} = J \begin{bmatrix} w_t - w^\star \\ w_{t-1} - w^\star \end{bmatrix} \quad \text{with} \quad J = \begin{bmatrix} (1+\beta)I_d - hA(I_d - \gamma A) & -\beta I_d \\ I_d & 0_d \end{bmatrix}, \tag{58}$$

where $I_d$ and $0_d$ respectively stand for the identity and the null matrices. To show the local convergence of (restarted) MEG for non-affine vector fields in Theorem 9, we first establish the following lemma, which connects the augmented state and the non-augmented one.

**Lemma 3.** *Let $P_t^{MEG}$ be the residual polynomial associated with $t$ updates of MEG (c.f., Theorem 1). Let $J$ be defined as (58). If $w_1 = w_0 - \frac{h}{1+m}v(w_0 - \gamma v(w_0))$, we then have*

$$J^t \begin{bmatrix} w_1 - w^\star \\ w_0 - w^\star \end{bmatrix} = \begin{bmatrix} P_t^{MEG}(A)(w_1 - w^\star) \\ P_t^{MEG}(A)(w_0 - w^\star) \end{bmatrix}. \tag{59}$$

*Consequently, if we denote $z_{t+1} := [w_{t+1}, w_t]$ and $z_* := [w^\star, w^\star]$, we have*

$$\|z_{t+1} - z_*\| \leqslant C(t+1)(1-\varphi)^t \|z_0 - z_*\|. \tag{60}$$

*Proof.* Let us express $J^t$ such that

$$J^t = \begin{bmatrix} P_t^{11}(A) & P_t^{12}(A) \\ P_t^{21}(A) & P_t^{22}(A) \end{bmatrix}, \quad \text{and} \quad J^t \begin{bmatrix} w_1 - w^\star \\ w_0 - w^\star \end{bmatrix} = \begin{bmatrix} P_t^{11}(A)(w_0 - w^\star) + P_t^{12}(A)(w_0 - w^\star) \\ P_t^{21}(A)(w_0 - w^\star) + P_t^{22}(A)(w_0 - w^\star) \end{bmatrix}. \tag{61}$$

By writing $J^{t+1} = J J^t$ and using the block-matrix form of $J$ in (58), we get that for any $t \geqslant 0$,

$$P_{t+1}^{11}(A) = ((1+\beta)I_d - hA(I_d - \gamma A))P_t^{11}(A) - \beta P_t^{21}(A)$$
$$P_{t+1}^{21}(A) = P_t^{11}(A) \tag{62}$$
$$P_{t+1}^{12}(A) = ((1+\beta)I_d - hA(I_d - \gamma A))P_t^{12}(A) - \beta P_t^{22}(A)$$
$$P_{t+1}^{22}(A) = P_t^{12}(A). \tag{63}$$

Hence, we have that,

$$P_{t+1}^{11}(A) \overset{(62)}{=} ((1+\beta)I_d - hA(I_d - \gamma A))P_t^{11}(A) - \beta P_{t-1}^{11}(A) \tag{64}$$

$$P_{t+1}^{12}(A) \overset{(63)}{=} ((1+\beta)I_d - hA(I_d - \gamma A))P_t^{12}(A) - \beta P_{t-1}^{12}(A). \tag{65}$$

We claim that

$$P_t^{11}(A) + P_t^{12}(A) = P_t^{\text{MEG}}(A) \quad \text{for all} \quad t \geqslant 0. \tag{66}$$

We prove this via induction.

For the base case, using the fact that $w_1 = w_0 - \frac{h}{1+m}(w_0 - \gamma v(w_0))$, we have that

$$(P_1^{11}(A) + P_1^{12}(A))(w_0 - w^\star) = w_1 - w^\star = (I_d - \frac{h}{1+m}(I_d - \gamma A))(w_0 - w^\star) = P_1^{\text{MEG}}(A)(w_0 - w^\star),$$
$$(P_0^{11}(A) + P_0^{12}(A))(w_0 - w^\star) = (P_1^{21}(A) + P_1^{22}(A))(w_0 - w^\star) = I_d(w_0 - w^\star) = P_0^{\text{MEG}}(A)(w_0 - w^\star).$$

To show the induction step, by adding (64) and (65), we get

$$P_{t+1}^{11}(A) + P_{t+1}^{12}(A) = ((1+\beta)I_d - hA(I_d - \gamma A))(P_t^{11}(A) + P_t^{12}(A)) - \beta(P_{t-1}^{11}(A) + P_{t-1}^{12}(A))$$
$$\overset{(66)}{=} ((1+\beta)I_d - hA(I_d - \gamma A))P_t^{\text{MEG}}(A) - \beta P_{t-1}^{\text{MEG}}(A),$$

where in the last step we used the induction hypothesis. Also notice $P_{t+1}^{11}(A) + P_{t+1}^{12}(A) = P_{t+1}^{\text{MEG}}$ on the left-hand side.

Hence we have for any $t \geqslant 0$,

$$(P_t^{11}(A) + P_t^{12}(A))(w_0 - w^\star) = P_t^{\text{MEG}}(A)(w_0 - w^\star). \tag{67}$$

Therfore, going back to (61), we have:

$$\begin{bmatrix} w_{t+1} - w^\star \\ w_t - w^\star \end{bmatrix} = J^t \begin{bmatrix} w_1 - w^\star \\ w_0 - w^\star \end{bmatrix} = \begin{bmatrix} (P_t^{11}(A) + P_t^{12}(A))(w_0 - w^\star) \\ (P_t^{21}(A) + P_t^{22}(A))(w_0 - w^\star) \end{bmatrix}$$
$$\overset{(62),(63)}{=} \begin{bmatrix} (P_t^{11}(A) + P_t^{12}(A))(w_0 - w^\star) \\ (P_{t-1}^{11}(A) + P_{t-1}^{12}(A))(w_0 - w^\star) \end{bmatrix}$$
$$\overset{(66)}{=} \begin{bmatrix} P_t^{\text{MEG}}(A)(w_0 - w^\star) \\ P_{t-1}^{\text{MEG}}(A)(w_0 - w^\star) \end{bmatrix}$$
$$\overset{\text{Thm. } 1}{=} \begin{bmatrix} P_t^{\text{MEG}}(A)(w_0 - w^\star) \\ P_t^{\text{MEG}}(A)(w_{-1} - w^\star) \end{bmatrix}$$
$$= \begin{bmatrix} P_t^{\text{MEG}}(A) & 0 \\ 0 & P_t^{\text{MEG}}(A) \end{bmatrix} \cdot \begin{bmatrix} w_0 - w^\star \\ w_{-1} - w^\star \end{bmatrix}$$
$$= P_t^{\text{MEG}}(A) \otimes I_2 \cdot \begin{bmatrix} w_0 - w^\star \\ w_{-1} - w^\star \end{bmatrix},$$

where we use the convention that $w_0 = w_{-1}$. Finally, using the fact that $\|A \otimes B\| = \|A\|\|B\|$ for $\ell_2$-operator norm (Lancaster & Farahat, 1972), we have

$$\|z_{t+1} - z_*\| \leqslant \|P_t^{\text{MEG}}(A)\|\|z_0 - z_*\| \overset{(10)}{\leqslant} C(t+1)(1-\varphi)^t\|z_0 - z_*\|.$$

$\square$

### D.1 Proof of Theorem 9

*Proof.* We first recall the restarted MEG algorithm we consider in (31):

$$[w_{tk+i+1}, w_{tk+i}] = G([w_{tk+i}, w_{tk+i-1}]) \quad \text{for} \quad 1 \leqslant i \leqslant k-1, \quad \text{and then}$$
$$w_{(t+1)k+1} = w_{(t+1)k} - \frac{h}{1+m}v(w_{(t+1)k} - \gamma v(w_{(t+1)k})).$$

In other words, we repeat MEG for $k$ steps, and then re-start the momentum at $[w_{(t+1)k+1}, w_{(t+1)k}]$.

We can analyze this method as follows, where we denote $z_t := [w_t, w_{t-1}]$ and $z_* = [w^\star, w^\star]$:

$$\begin{aligned}
\|z_{(t+1)k} - z_*\| &= \|G^{(k)}(z_{tk}) - z_*\| \\
&\overset{(68)}{=} \|\nabla G^{(k)}(\tilde{z}_{tk})(z_{tk} - z_*)\| \\
&\leqslant \|\nabla G^{(k)}(z_*)(z_{tk} - z_*)\| + \|(\nabla G^{(k)}(\tilde{z}_{tk}) - \nabla G^{(k)}(z_*))(z_{tk} - z_*)\| \\
&\overset{(60)}{\leqslant} C(k+1)(1-\varphi)^k\|z_{tk} - z_*\| + \|\nabla G^{(k)}(\tilde{z}_{tk}) - \nabla G^{(k)}(z_*)\|\|(z_{tk} - z_*)\|,
\end{aligned}$$

where in the second line we use the Mean Value Theorem:

$$\begin{aligned}
\exists \tilde{z}_{tk} \in [z_{tk}, z_*] \quad \text{such that} \quad G^{(k)}(z_{tk}) &= G^{(k)}(z_*) + \nabla G^{(k)}(\tilde{z}_{tk})(z_{tk} - z_*) \\
&= z_* + \nabla G^{(k)}(\tilde{z}_{tk})(z_{tk} - z_*) \quad \text{(since $z_*$ is the fixed point).} \quad (68)
\end{aligned}$$

In the fourth line we used the fact that $\nabla G^{(k)}(z_*)(z_{tk} - z_*)$ exactly correspond to $k$ updates of MEG when the vector field is affine, as well as Lemma 3 to account for the augmented state.

Now let us consider $\varphi > \varepsilon > 0$ and $k$ large enough such that $C(k+1)(1-\varphi)^k \leqslant (1-\varphi+\frac{\varepsilon}{2})^k$. Since $\nabla G$ is assumed to be continuous, $\nabla G^{(k)}$ is continuous too. Therefore, there exists $\delta > 0$ such that $\|z_{tk} - z_*\| \leqslant \delta$ implies $\|\nabla G^{(k)}(\tilde{z}_{tk}) - \nabla G^{(k)}(z_*)\| \leqslant \varepsilon'$. In particular, choose $\varepsilon' = (1-\varphi+\varepsilon)^k - (1-\varphi+\frac{\varepsilon}{2})^k \sim \frac{k\varepsilon}{2(1-\varphi)}$. Then, we have

$$\begin{aligned}
\|z_{(t+1)k} - z_*\| &\leqslant C(k+1)(1-\varphi)^k\|z_{tk} - z_*\| + \|\nabla G^{(k)}(\tilde{z}_{tk}) - \nabla G^{(k)}(z_*)\|\|z_{tk} - z_*\| \\
&\leqslant (1-\varphi+\tfrac{\varepsilon}{2})^k\|z_{tk} - z_*\| + \varepsilon'\|z_{tk} - z_*\| \\
&\leqslant (1-\varphi+\varepsilon)^k\|z_{tk} - z_*\| < \|z_{tk} - z_*\| < \|z_0 - z_*\|.
\end{aligned}$$

From the above, we can conclude that for all $\varepsilon > 0$, there exists $k > 0$ and $\delta > 0$ such that, for all initialization satisfying $\|w_0 - w^\star\| \leqslant \delta$, the restarted MEG described above satisfies:

$$\|w_t - w^\star\| = O((1-\varphi+\varepsilon)^t)\|w_0 - w^\star\|.$$

$\square$

