# OpenReview forum: "When is Momentum Extragradient Optimal? A Polynomial-Based Analysis"
_TMLR — Accepted by TMLR_

### Review · Reviewer_h1nW · 2023-09-26

**Summary Of Contributions:**

The authors consider the extra gradient method with momentum (MEG). They investigate how the spectrum of a Jacobian has an effect on the method. A detailed analysis and the choice of hyperparameters to get acceleration are provided.

**Audience:**

Yes

**Broader Impact Concerns:**

-

**Claims And Evidence:**

Yes

**Requested Changes:**

-

**Strengths And Weaknesses:**

I will be very brief because I do not have huge expertise in the extra gradient method. In general, I think that the work is solid and considered an interesting problem. The authors carefully explored different spectrum regimes when MEG converges and outperforms GD and EG.

Some weaknesses:
1. As far as I understand, the whole provided theory only works with *affine functions*. The authors should more clearly emphasize it in the abstract and in the introduction.
2. Moreover, the authors show that MEG enjoys acceleration in Theorem 4 and compare this result to GDM. While I agree with this result, I think that this comparison is not fair because GDM can obtain acceleration even with *non-affine gradients*. I believe that this should be explained. The same comment should be addressed when the authors compare MEG with other previous methods.

I think the paper is good and deserves acceptance, but the authors should more clearly explain that they consider the setup where the objective is an *affine function*.

---

> ### Author Response · Authors · 2023-11-14
> **Thank you for the review**
>
> Dear reviewer h1nW,
>
> Thank you for your comments and thoughtful feedback. We are encouraged you found our work interesting and deserves acceptance. We hope our reply below clarifies some of your concerns.
>
> **Q: The whole provided theory only works with affine functions. The authors should more clearly emphasize it in the abstract and in the introduction.**
>
> **A**: Thank you for the comment. We can more clearly emphasize in the abstract and in the introduction that the provided theory applies when the vector field is affine. *Please note, however, that we also extended the theory: we prove the local convergence of MEG for non-affine vector fields in the Appendix (highlighted in blue in the supplementary material), by considering restarting the momentum of MEG after $k$ iterations.* We believe this additional result strengthens our paper and addresses the concern that the provided theory only applies to affine vector fields.
>
> **Q: Comparison with GDM does not seem fair.**
>
> **A**: Thank you for the thoughtful question. The result in Theorem 4 is twofold: MEG not only achieves an accelerated convergence rate $(1 - O(\sqrt{\tau}))$, but also enjoys constant level improvement $(\sqrt{1-R^2})$ compared to GDM, as seen in Eq. (18). As we write in the paragraph below Eq. (18), achieving this improvement is possible by having additional information beyond just the largest ($L_2$) and the smallest ($\mu_1$) eigenvalues of the Hessian. We believe this is a fair comparison because we compare GDM and MEG with the same game Jacobian spectrum. Finally, it was recently shown that GDM does not achieve acceleration beyond quadratic minimization (i.e., affine vector field) [1].
>
> We hope the above answers clarify your concern. Thank you again for your time in reviewing our manuscript.
>
> [1] B. Gouja,d A. Taylor, and A. Dieuleveut, “Provable non-accelerations of the heavy-ball method”

---

### Review · Reviewer_uZGK · 2023-09-29

**Summary Of Contributions:**

In this work, the authors demonstrate the benefit of momentum extragradient (MEG) in terms of convergence for bilinear games over gradient descent (GD), extragradient (EG) and GDM (Gradient descent with momentum) methods. The authors assume the game vector field to be an affine function and analyses the convergence of these algorithms in 3 sub-cases where the game Jacobian takes only 1) real eigenvalues, 2) cross spectrum and 3) shifted imaginary spectrum. They showcase the accelaration of MEG by considering the optimal hyperparameters (step-size, extragradient parameter and momentum parameter) that result in the best convergence.

**Audience:**

Yes

**Broader Impact Concerns:**

No ethical concerns

**Claims And Evidence:**

Yes

**Requested Changes:**

I would like to see some comments on the 4 weakness points i mentioned. For point 4, the implicit regualrization won't affect convergence on a quadratic games but can have a different effect for more complex games.

I feel addressing point number 1 and 2 are critical in securing my recommendation for acceptance.

**Strengths And Weaknesses:**

Strengths:
1) The problem considered here is important and the authors have made a significant contribution in this work.
2) The paper is easy to read and follow. The mathematical explanations are quite easy to follow as well.
3) The 3 case convergence analysis based on the spectra of the game Jacobian is rigorous. I did not find any mistakes in the derivation.

Weakness:
1) In my opinion, the assumption of the game vector field to be an affine function is too restrictive which makes this analysis valid for only quadratic games. I did not see the authors commenting about this in a more general setting where the game vector field is not affine and the polynomial based analysis will no longer be valid.

2) Is it possible to extend the setup of the experiments to explore beyond quadratic games and analyze the convergence of these 4 methods (GD,EG,MEG,GDM)?

3) Can the authors comment on how the convergence is affected with stochasticity in the gradients with all these cases? Currently only the deterministic setting is considered.

4) Although EG methods are known to perform better than GD, the use of finite small step-size has an implicit effect of discretization drift [1] , where the implicit regularization is exactly the norm of the gradient. Infact applying a first order taylor expansion on the extra-gradient term would also have this term ( the norm of the gradient). In this context, how important is having the explicit regularization (as extra gradient term) compare with the implcit regularization due to the finite step size? Morevoer, in this context, implicit regularization for momentum (due to finite step-size and momentum parameter) is already explored in [2] and whether this can be compared with MEG (momentum extragradient).


[1] Rosca, Mihaela C., et al. "Discretization drift in two-player games." International Conference on Machine Learning. PMLR, 2021.


[2] Ghosh, Avrajit, et al. "Implicit regularization in Heavy-ball momentum accelerated stochastic gradient descent." The Eleventh International Conference on Learning Representations. 2022.

---

> ### Author Response · Authors · 2023-11-14
> **Thank you for the review**
>
> Dear reviewer uZGK,
>
> Thank you for your valuable comments and feedback. We are encouraged you found our contribution significant, and the presentation of the paper is easy to follow. We hope our reply below clarifies some of your concerns.
>
> **Q: The whole provided theory only works with affine functions. What can be said about general vector field where polynomial based analysis no longer hold?**
>
> **A:** While it is true that the polynomial-based analysis is only valid for quadratic games, our theoretical result provides interesting insights as to how MEG behaves differently in the presence of convoluted vector field dynamics, as illustrated in Figure 1. To the best of our knowledge, a convergence of any algorithm in the “cross-shaped” spectrum or “shifted imaginary” spectrum has not been considered in the literature before; set aside obtaining the optimal hyperparameters of MEG we obtain in the paper.
>
> That being said, *we also extended the theory: we prove the local convergence of MEG for non-affine vector fields in the Appendix (highlighted in blue in the supplementary material) by considering restarting the momentum of MEG after $k$ iterations.* We believe this additional result strengthens our paper and addresses the concern that the provided theory only applies to affine vector fields.
>
> **Q: Is it possible to extend beyond quadratic games?**
>
> **A:** As mentioned earlier, we have an extended theory where we prove the local convergence of MEG for non-affine vector fields in the Appendix (highlighted in blue) by considering restarting the momentum of MEG after $k$ iterations. We believe this additional result strengthens and addresses the concern.
>
> **Q: How is the convergence affected with stochasticity?**
>
> **A:** Unfortunately, extending the polynomial-based theoretical analysis to incorporate stochasticity is non-trivial. Empirically, we remind the reviewer that one of the experiments considered in Section 6 does *not* follow the cross-shaped spectrum $\mathcal{S}_2^\star$ exactly. This experiment shows how the MEG with optimal hyperparameters performs with a perturbed spectrum, which can be roughly connected to perturbing gradients in the affine vector field case. There, it shows that MEG converges much faster than other first-order methods, even after performing grid-search.
>
> **Q: Connection to implicit regularization due to finite step size and/or due to momentum?**
>
> **A:** We thank the reviewer for the thoughtful questions. Indeed, applying a first-order Taylor expansion on the extragradient term would also have the norm of the gradient. However, studying the effect of the (explicit) extragradient term and how it interacts with the implicit regularization due to finite step size is out of the scope of this work. Further, as the reviewer noted, the momentum parameter of MEG can also interact with the implicit regularization in a highly non-trivial way. We thank the reviewer for the reference, and we will note this connection in the revision as a future research direction.
>
> We hope the above answers clarify your concern. Thank you again for your time in reviewing our manuscript.

---

### Review · Reviewer_4b8x · 2023-11-02

**Summary Of Contributions:**

This paper analyzes the momentum extragradient (MEG) method for differentiable games using residual polynomial.
It is shown that the convergence rate of the MAG can be characterized by the residual polynomial (Theorem 1),
under the assumption that the vector field is given by an affine function.
Based on this theorem,
the paper provides sufficient conditions for convergence and a guide to setting optimal parameters.
The usefulness of this analysis is confirmed by numerical experiments as well.

**Audience:**

Yes

**Claims And Evidence:**

Yes

**Requested Changes:**

- It would be better to mention to what extent the analytical results given for an affine problem (Theorems 1 and 2) can be extended to non-affine problems. Even if there is no clear theoretical argument and the application to non-affine settings is heuristics, that in itself is fine in my opinion. However, in that case, I think it is necessary to clarify that it is heuristics.
- I would prefer to see a clear statement of how this paper defines "optimal parameters" before theorems on optimal parameters given in Section 4.
- There are several places (e.g., (12), (14) and (15), ...) where the symbol of $\in$ is used in the sense of inclusion, i.e., in situations where the symbols $\subset$ or $\subseteq$ would normally be used. I would appreciate it if you could check them.

**Strengths And Weaknesses:**

Strengths:

- Motivation and contribution are well explained.
- The research topic of this paper, convergence analysis of dynamics for games, would be of interest to ML communities.
- The paper contains practical results, such as guidelines for optimal parameter settings.

Weaknesses:

- The main results of this paper are natural extensions of the results by Azizian et al. (2020a, 2020b) and do not appear to be a major breakthrough.
- It was not clear to me that the main theoretical results regarding convergence rates (Theorems 1 and 2) were applicable to the general games with non-affine vector fields.

Reference:
- Waïss Azizian, Ioannis Mitliagkas, Simon Lacoste-Julien, and Gauthier Gidel. A tight and unified analysis
of gradient-based methods for a whole spectrum of differentiable games. In International Conference on
Artificial Intelligence and Statistics, pp. 2863–2873. PMLR, 2020a.
- Waïss Azizian, Damien Scieur, Ioannis Mitliagkas, Simon Lacoste-Julien, and Gauthier Gidel. Accelerating
smooth games by manipulating spectral shapes. In International Conference on Artificial Intelligence and
Statistics, pp. 1705–1715. PMLR, 2020b

---

> ### Author Response · Authors · 2023-11-14
> **Thank you for the review**
>
> Dear reviewer 4b8x,
>
> Thank you for the comments and your valuable feedback. We are glad you find the motivation and contribution of our paper clear and the convergence analysis and guidelines for optimal hyperparameters to be of interest to ML communities. We hope our answers below clarify some of your concerns.
>
> **Q: Natural extensions of the results by Azizian et al. (2020a, 2020b)?**
>
> **A:** Thank you for the comment. Our results are not necessarily natural extensions of the Azizian et al. (2020a, 2020b). Firstly, please note that Azizian et al. (2020a) take a vastly different approach than ours; they consider the setting where the eigenvalues of the jacobian of the game vector field at the optimum are assumed to have all positive real parts. As a result, the rate they achieve does not lead to an accelerated one for all three spectrum models considered, as we show theoretically in Corollary 1 and experimentally in Section 6. They also rely on spectral analysis of operators in terms of analysis technique, which is different from the polynomial-based analysis we take.
>
> Azizian et al. (2020b) are more closely related to our work, but their setting is simpler; they mainly consider bilinear games, which have *purely imaginary eigenvalues*. Hence, they can show an accelerated result of GDM for bilinear games by reducing them to games with real eigenvalues (i.e., minimization), and naturally, GDM achieves an accelerated convergence rate as it does for quadratic minimization, the only setting GDM can achieve acceleration [1]. Such a reduction is possible since bilinear games have a simple spectrum. On the other hand, our scenario is more convoluted, as shown in Figure 2 (e.g., when real and complex eigenvalues co-exist), and showing acceleration, set aside obtaining optimal hyperparameters, is nontrivial. This is confirmed by Corollary 2 in our paper, where the results of Azizian et al. (2020b) do not lead to an acceleration when the condition number exceeds $\sqrt{5}$.
>
> **Q: It was not clear to me that the main theoretical results regarding convergence rates (Theorems 1 and 2) were applicable to the general games with non-affine vector fields.**
>
> **A:** It is nontrivial to extend the polynomial-based analysis to non-affine vector fields. *Please note, however, that we extended the theory: we prove the local convergence of MEG for non-affine vector fields in the Appendix (highlighted in blue) by considering restarting the momentum of MEG after $k$ iterations.* We believe this additional result strengthens our paper and addresses the concern that the provided theory only applies to affine vector fields.
>
> **Q: I would prefer to see a clear statement of how this paper defines "optimal parameters" before theorems on optimal parameters given in Section 4.**
>
> **A:** Thank you for the suggestion. The optimal parameters are the step sizes and the momentum parameters that minimize the asymptotic convergence rate in Theorem 2, given each of the three cases of Jacobian spectrum models. In the supplementary material, we added this clarification after Theorem 2 (in blue).
>
> **Q: Several places where the symbol is not used correctly**
>
> **A:** Thank you for the correction. We have updated the supplementary material accordingly.
>
> We hope the above answers clarify your concern. Thank you again for your time in reviewing our manuscript.
>
> [1] B. Gouja,d A. Taylor, and A. Dieuleveut, "Provable non-accelerations of the heavy-ball method"

---

### Decision · Action_Editor_WkYg · 2023-12-11

**Recommendation:** Accept as is

**Comment:**

The primary concerns raised by the reviewers were (i) the difference from the previous result [Azizian et al. (2020a, 2020b)] and (ii) the restriction of the problem that assumes the affine vector field. The authors have addressed these concerns well in the rebuttal and the revision. Specifically, the authors clearly described the originality of the work and extended the theory into the non-affine case with the local convergence analysis. Therefore, the paper meets the acceptance criteria.

**Audience:**

The paper that studies convergence of  optimization dynamics for games would be of interest to TMLR audiences.

**Claims And Evidence:**

This work showed momentum extragradient (MEG) for differentiable games achieves accelerated convergence rates under specific setups, over GD, EG, and GD with momentum. The claim is theoretically well supported by clarifying three cases of eigenvalue distributions of Jacobian to achieve acceleration; eigenvalues are distributed: (i) on the (positive) real line, (ii) both on the real line along with complex conjugates, and (iii) only as complex conjugates.